# On the rankability of visual embeddings

**Ankit Sonthalia**
University of Tübingen
ankit.sonthalia@uni-tuebingen.de

**Arnas Uselis**
University of Tübingen
arnas.uselis@uni-tuebingen.de

**Seong Joon Oh**
University of Tübingen

## Abstract

We study whether visual embedding models capture continuous, ordinal attributes along linear directions, which we term *rank axes*. We define a model as *rankable* for an attribute if projecting embeddings onto such an axis preserves the attribute's order. Across 7 popular encoders and 9 datasets with attributes like age, crowd count, head pose, aesthetics, and recency, we find that many embeddings are inherently rankable. Surprisingly, a small number of samples, or even just two extreme examples, often suffice to recover meaningful rank axes, without full-scale supervision. These findings open up new use cases for image ranking in vector databases and motivate further study into the structure and learning of rankable embeddings. Our code is available at https://github.com/aktsonthalia/rankable-vision-embeddings.

## 1 Introduction

Visual embeddings are widely used for image retrieval. This relies on embeddings forming a metric space, where similar images are placed nearby. Modern visual encoders generally satisfy this property, and many systems depend on it in the form of vector databases.

Ranking is another core operation in databases. It allows users to navigate large collections by sorting, paginating, and filtering results. For instance, e-commerce platforms like Amazon benefit from ranking product images by visual quality or certain product-specific attributes (*e.g.,* ranking shoes by how formal they look).

In this work, we ask: **are visual embeddings also rankable?** Retrieval relies on *local* similarity around a query. Ranking requires a *global* ordering along an attribute. Prior work has largely addressed the former. The global rankability of embeddings remains underexplored.

**Retrieval**      **Ranking**

We define **rankability** as follows: given an embedding function $f$ and a continuous attribute $A$ (*e.g.,* "age"), we say $f$ is rankable with respect to $A$ if there exists a **rank axis** $v_A$ such that the projection $v_A^\top f(x)$ preserves the correct order of the target attribute $A(x)$ over a dataset. For instance, if $A$ denotes "age", this projection would sort face images from youngest to oldest.

We examine two questions: (1) Are visual embeddings rankable? (2) How easily can we recover the rank axis for a given attribute?

39th Conference on Neural Information Processing Systems (NeurIPS 2025).

To address (1), we evaluate 7 modern visual encoders, from ResNet to CLIP, across 9 datasets with 7 attributes: age, crowd count, 3 head pose angles (pitch, roll, yaw), image aesthetics, and recency. We find that many embedding spaces are indeed rankable (Section 3).

To address (2), we estimate the rank axis $v_A$ with minimal supervision. The structure of the embedding space makes full-dataset regression unnecessary. In many cases, a handful of annotated samples and, in some cases, a pair of samples $x_l$ (low) and $x_h$ (high) already recover non-trivial ranking performance. For the latter case, we define the rank axis as $v_A = (f(x_h) - f(x_l))/\|f(x_h) - f(x_l)\|_2$. This opens up the possibility for fast ordering of new images by arbitrary attributes. For example, a photo app lets users sort selfies by age appearance. It uses CLIP embeddings and two reference images: one of a child and one of an elderly person. The app computes $v_{\text{age}}$ without training. Users scroll from youngest-looking to oldest-looking faces in their album (Section 4).

Contributions:

1. We define and motivate rankability as a property of visual embeddings, distinct from retrieval.

2. We study rankability across modern encoders and real-valued attributes; results show that current embeddings are rankable.

3. We show that rank axes can sometimes be recovered using only two or a handful of labelled samples.

## 2    Related work

**Embeddings for retrieval.** Visual encoders are commonly used to index images in vector databases, enabling nearest neighbour search for retrieval tasks [42, 52, 33, 57]. This setup, known as deep metric learning [5, 6, 42], predates vision-language models like CLIP [52]. CLIP and related models shifted the focus to *cross-modal* similarity modelling, where vision and language share a joint embedding space used for classification [52], retrieval [75, 2], and captioning [39, 26]. While the majority of work in visual encoders is devoted to the understanding of the local similarity structure, we study how visual embeddings support *global* ranking instead of just local retrieval.

**Improving embedding geometry and structure.** Prior work has explored ways to improve the geometry of the embedding space. Order embeddings and hyperbolic representations have been used to model hierarchies [65, 21, 8, 51]. Training disentangled representation [71] is considered critical for compositionality, where attributes are assigned to certain linear subspaces [60, 3]. Others have defined concepts like uniformity and separability of the representations [70]. In this work, we focus on the analysis of a wide range of visual encoders, rather than introducing recipes for improvements.

**Analysing embedding geometry and structure.** A large body of work has examined the geometry and structure of CLIP's learned embedding space. CLIP and its derivatives have been studied extensively [4, 53, 32, 77]. Several works have reported modality gaps between vision and language embeddings [12]. Some studies point to the absence of certain structures and capabilities in CLIP representations: attribute-object bindings [31, 80, 25], or the association of attributes to corresponding instances. Others argue that much information is already present in CLIP representations, including parts-of-speech and linguistic structure [44], attribute-object bindings [24], and compositional attributes [62, 63]. The platonic representation hypothesis further suggests that models converge to similar internal structures [19]. In this work, we analyse the embedding geometry and structure for modern visual embeddings from the novel perspective of rankability.

**Linearly probing an embedding.** Linear probing is a fast and widely used method to test for the presence of concepts in visual embeddings [23, 18, 64]. It measures the accuracy of a linear classifier trained on intermediate-layer features, effectively testing whether a hyperplane can separate embeddings containing a concept from those that do not. This technique has been used to study the geometry of CLIP's embeddings [30] and to probe for specific information such as attribute-object bindings and compositionality [24, 62]. While effective for binary separability, linear probes are limited in capturing non-binary, ordinal, or relational structures [1]. Prior work has not directly analysed how *continuous* attributes are laid out in the embedding space. Our work extends this line of research by moving beyond concept detection to characterise the internal structure of embeddings along ordinal axes, introducing rankability as a new property not captured by prior probing methods.

**Ordinal information in embeddings.** Early works on relative and ordinal attributes explored how continuous visual attributes could be inferred and ranked [50, 36, 84, 14, 76]. However, these efforts were limited to smaller models and datasets. Recent studies have begun to examine ordinal signals in large pre-trained models. These include aligning CLIP with ordinal supervision [69], and applying CLIP or general VLMs to specific tasks such as aesthetics [74, 67], object counting [48, 20], crowd counting [35], and difference detection [56, 22]. Several works have adapted CLIP for ranking through prompt tuning [34], adapter-based methods [79] or regression-based fine-tuning [10, 68]. Despite these efforts, most focus on task-specific performance or modifying the embedding space, rather than understanding its internal ordinal structure in the embeddings themselves. Our work fills this gap by systematically analysing the rankable structure of existing visual encoders and revealing the presence of ordinal directions in their embedding spaces.

## 3 Are vision embeddings rankable?

We set out to answer the question. For this, we first formally define rankability and strategies to measure it (Section 3.1). We introduce the models and data used for our experiments in Section 3.2. We present results in Section 3.3.

**Notation.** Our experiments use RGB image datasets with real-valued attribute labels. We denote an image dataset as $X \subset \mathbb{R}^{3 \times H \times W}$ and define an *ordinal attribute A* as a function $A : X \to Y \subset \mathbb{R}$ where $Y$ is the range of possible labels and $A(x)$ is the ground-truth label for a given image $x$. An image encoder is a function $f : X \to \mathbb{R}^d$ where $d$ is the dimensionality of the embedding space. We occasionally use the term "representation" to refer to an image encoder.

### 3.1 Rankability

We aim to characterize the *linear* structure of the ordinal information present in visual embedding spaces. Our definition of rankability then naturally emerges as follows.

**Definition 1** (Rankability). *A representation $f : X \to \mathbb{R}^d$ is **rankable** for an ordinal attribute $A$ over an image dataset $X$ if there exists a **rank axis** $v_A \in \mathcal{S}^{d-1}$, a $d-1$ dimensional unit sphere, such that for any $x_1, x_2 \in X$ with $A(x_1) \geq A(x_2)$, it follows that $v_A^\top f(x_1) \geq v_A^\top f(x_2)$.*

The above definition requires a rank axis $v_A$ to exactly preserve the ordering provided by the attribute $A$ over the dataset $X$. In practice, we measure rankability using the generalisation performance of the rank axis $v_A$ learned on a training split $X_{\text{train}}$ and tested on a disjoint split $X_{\text{test}}$. In this section, we obtain $v_A$ via linear regression on the labelled samples: $\{(f(x_i), a_i)\}_i$, where $a_i \in \mathbb{R}$ is the ground-truth continuous attribute label for each $x_i$. In Section 4, we consider approaches that do not require access to the attribute labels $a_i$.

**Spearman's rank correlation coefficient (SRCC)**, denoted as $\rho$ serves as our primary metric quantifying the monotonicity of the relationship between the true ordinality of the attribute $A$ and the predicted ranking along $v_A$. SRCC is widely used in related contexts [78].

We provide three reference points for the obtained rankability to contextualise the SRCC values:

**(1) No-train ("lower bound").** Even for untrained visual encoders, the embeddings do not result in null rank correlation ($\rho$=0). In order to correctly capture the no-information baseline, we consider the performances of randomly initialized versions of each encoder considered [61]. The optimal rank axis $v_A$ in this space serves as a lower bound for the rankability of the pretrained encoder.

**(2) Nonlinear (upper bound for embedding).** To estimate the total ordinal information in the given embedding, we use a two-layer multilayer perceptron (MLP), known to be a universal approximator [16]. Comparing against a non-linear regressor lets us estimate the proportion of ordinal information in an embedding that can be extracted linearly with a rank axis.

**(3) Finetuning (upper bound for encoder architecture).** To measure a broader upper bound indicating the capacity of the encoder architecture and the learnability of the attribute, we finetune the encoder. This conceptually envelops the nonlinear regression upper bound.

## 3.2 Experimental details

We provide further details on the list of attributes, datasets, encoder architectures, and the model selection protocol.

### 3.2.1 Attributes and datasets

In total, we use 9 datasets, covering 7 attributes. We provide a detailed breakdown in Table 1.

Table 1: **Datasets and attributes**. Summary of datasets used for evaluating the rankability of visual representations.

| Attribute | Dataset | #Train-val | #Test | Label Range | Split |
|---|---|---|---|---|---|
| Age | UTKFace [83] | 13,146 | 3,287 | 21–60 | From [27] |
| | Adience [11] | ~14k | ~4k | 8 age groups | Official 5-fold |
| Crowd count | UCF-QNRF [7] | 1,201 | 334 | 49–12,865 | Official |
| | ShanghaiTech-A [82] | 300 | 182 | 33–3,139 | Official |
| | ShanghaiTech-B [82] | 400 | 316 | 9–578 | Official |
| Pitch | BIWI Kinect [13] | 10,493 | 4,531 | ±60° | 6 test seqs. |
| Yaw | BIWI Kinect | 10,493 | 4,531 | ±75° | 6 test seqs. |
| Roll | BIWI Kinect | 10,493 | 4,531 | ±45° | 6 test seqs. |
| Aesthetics | AVA [41] | 230,686 | 4,692 | Ratings (1–10) | From [78] |
| | KonIQ-10k [17] | 8,058 | 2,015 | Ratings (1–100) | Official |
| Image modernness | Historical Color Images [49] | 1,060 | 265 | 5 decades | From [78] |

### 3.2.2 Architectures

We test representative image-only and CLIP-based encoders. Among image-only encoders, we use the ResNet50 [15], ViT-B/32@224px [9], and ConvNeXtv2-L [73] architectures. Likewise, among CLIP encoders, we test the ResNet50, ViT-B/32, and ConvNeXt-L@320px variants. We also test DINOv2 [46] (ViT-B/14 variant). See Table 2 for more information.

Table 2: **Summary of vision encoders used in our study.** "Acronym" refers to how each model is denoted in our main results tables.

| Acronym | Architecture | #Dims | Year | #Params | Type | Input Size |
|---|---|---|---|---|---|---|
| RN50 | ResNet50 | 2048 | 2015 | 25.6M | ConvNet | 224×224 |
| ViTB32 | ViT-B/32 | 768 | 2020 | 88.2M | Transformer | 224×224 |
| CNX | ConvNeXtV2 | 1536 | 2023 | 198M | ConvNet | 224×224 |
| DINO-B14 | DINOv2 (ViT-B/14) | 768 | 2023 | 86.6M | Transformer | 518×518 |
| CLIP-RN50 | CLIP ResNet50 | 1024 | 2021 | 38.3M | ConvNet | 224×224 |
| CLIP-ViTB32 | CLIP ViT-B/32 | 512 | 2021 | 87.8M | Transformer | 224×224 |
| CLIP-CNX | CLIP ConvNeXtV2 | 768 | 2023 | 199.8M | ConvNet | 320×320 |

### 3.2.3 Hyperparameter tuning and model selection

Hyperparameters for reporting final performances on the test set are selected based on the best validation SRCC, using either official validation splits or holding them out from the corresponding training splits.

**Linear and nonlinear regression.** We test 30 random hyperparameter configurations per dataset-model pair. The initial learning rate is sampled from a log-uniform distribution over $[10^{-6},\ 10^{-1}]$ and decayed over a cosine schedule to zero, while the weight decay is sampled from a log-uniform distribution over $[10^{-7}, 10^{-4}]$. Data augmentation (horizontal flipping) is also toggled on or off randomly for a given run. We use 100 epochs and a batch size of 128 throughout. For nonlinear regression, we use a 2-layer MLP with 128 hidden dimensions and ReLU non-linearity.

**Finetuning.** For image-only ResNet50 and ConvNeXt encoders, we conduct a grid search over two encoder learning rates $(10^{-4}, 10^{-3})$ and two weight-decay rates $(0, 10^{-5})$. For ViT-B/32, DINOv2 and CLIP models, we conduct a larger grid search over three encoder learning rates $(10^{-7}, 10^{-6}, 10^{-5})$ and four weight-decay values $(0, 10^{-5}, 10^{-4}, 10^{-3})$. The downstream model always uses a learning rate of $0.1$. We use $20$ epochs for all runs. Batch size is fixed at $128$, except for ConvNeXt-v2, DINOv2 and CLIP-ConvNeXt-v2, where batch size is reduced to $16$ because of memory constraints. We use horizontal-flip augmentation in all finetuning runs to mitigate overfitting.

### 3.2.4 Compute resources

All experiments were conducted on a single NVIDIA A100 GPU with 40GB of memory. All primary experiments using frozen embeddings completed within 1–2 minutes, as we cache the embeddings. Finetuning experiments took between 1 and 24 hours each, depending on the dataset. In total, our experiments amounted to approximately 150 compute-days. We used the "stuned" Python library [55] for managing the experiments.

### 3.3 Results

**Setup.** The SRCC for the linear regressor measures the rankability of the underlying vision encoder, while the baselines (no-encoder, nonlinear regression and finetuning) provide reference points. We report our main results in Table 3 and Table 4. In Table 3, we average metrics across all architectures. Then in Table 4, we zoom into individual architectures while retaining only the rankability metric and averaging across all datasets that contain the same attribute. We also report qualitative results in Figure 1. Our observations vary across individual attributes and architectures, but some patterns emerge.

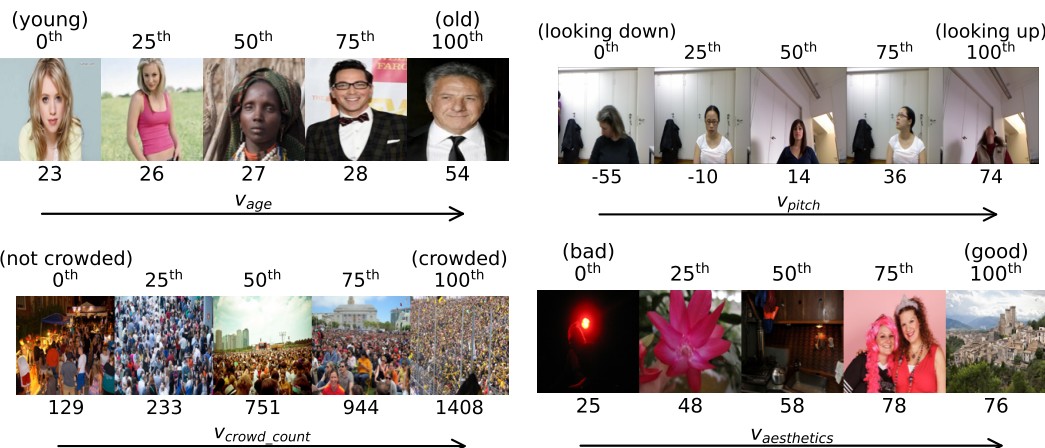

Figure 1: **Visualisation of rank axes.** We show $r^{\text{th}}$ percentile samples along the rank axes found using linear regression over CLIP-ViT-B/32 embeddings from each respective dataset.

**Average rankability of vision embeddings is non-trivially high.** We observe in Table 3 that with respect to age on the Adience dataset, the average rankability of $0.861$ is much higher than the no-train lower bound of $0.266$, while the nonlinear and finetuned upper bounds ($0.878$ and $0.910$) are only slightly higher in comparison. In fact, for all attributes except for yaw and roll, the average rankability is closer to the two upper bounds than to the lower bound.

**Linear regression is often at par with dedicated SOTA efforts.** Comparison with SOTA methods is outside of our main scope. We nevertheless find it interesting that often, linear regression over pretrained unmodified embeddings performs at par with dedicated SOTA efforts that require changing the underlying encoder or embeddings, or applying complex downstream models or specialised loss functions for the ranking task. For instance, as noted in Appendix E, on age estimation over UTKFace, linear regression achieves an MAE of $4.25$, only slightly underperforming [78] (MAE = $3.83$) and at par with [27] (MAE = $4.23$). Our results suggest that the rankability of visual embeddings provides a

Table 3: **Vision embeddings are generally rankable**. Spearman rank correlation $\rho$ between the true ranking of data samples and the predicted ranks. Higher is better. Results are averaged across all 7 architectures. **No-train**: Find $v_A$ on the embeddings of a randomly-initialised encoder. **Rankability**: How well does $v_A$ encode ordinal information? **Nonlinear**: Maximal non-linear ordinal information contained in embeddings. **Finetuned**: How learnable is the target attribute? For more information, see Section 3.1.

| Attribute (Dataset) | No-train lower bound | Rankability main | Nonlinear upper bound | Finetuned upper bound |
|---|---|---|---|---|
| Age (UTKFace) | 0.199 | 0.766 | 0.776 | 0.799 |
| Age (Adience) | 0.266 | 0.861 | 0.878 | 0.910 |
| Crowd (UCF-QNRF) | 0.220 | 0.843 | 0.854 | 0.886 |
| Crowd Count (ST-A) | 0.120 | 0.734 | 0.749 | 0.689 |
| Crowd Count (ST-B) | 0.135 | 0.869 | 0.887 | 0.840 |
| Pitch (Kinect) | 0.405 | 0.803 | 0.811 | 0.975 |
| Yaw (Kinect) | 0.078 | 0.434 | 0.597 | 0.967 |
| Roll (Kinect) | 0.151 | 0.218 | 0.326 | 0.859 |
| Aesthetics (AVA) | 0.156 | 0.653 | 0.692 | 0.693 |
| Aesthetics (KonIQ-10k) | 0.435 | 0.761 | 0.793 | 0.901 |
| Recency (HCI) | 0.324 | 0.680 | 0.688 | 0.722 |

Table 4: **Rankability across datasets and models**. Spearman's rank correlation $\rho$ between the true ranking of data samples and the predicted ranks. Higher is better. See Table 2 for model details.

| Attribute (Dataset) | Model | | | | | | |
|---|---|---|---|---|---|---|---|
| | RN50 | ViTB32 | CNX | DINO-B14 | CLIP-RN50 | CLIP-ViTB32 | CLIP-CNX |
| Age (UTKFace) | 0.633 | 0.739 | 0.772 | 0.770 | 0.820 | 0.810 | 0.820 |
| Age (Adience) | 0.723 | 0.828 | 0.871 | 0.853 | 0.898 | 0.924 | 0.928 |
| Crowd (UCF-QNRF) | 0.864 | 0.837 | 0.810 | 0.788 | 0.870 | 0.870 | 0.860 |
| Crowd (ST-A) | 0.799 | 0.700 | 0.695 | 0.653 | 0.760 | 0.750 | 0.780 |
| Crowd (ST-B) | 0.879 | 0.878 | 0.867 | 0.821 | 0.890 | 0.860 | 0.890 |
| Pitch (Kinect) | 0.663 | 0.673 | 0.909 | 0.716 | 0.860 | 0.920 | 0.880 |
| Yaw (Kinect) | 0.624 | 0.305 | 0.384 | 0.804 | 0.120 | 0.360 | 0.440 |
| Roll (Kinect) | 0.352 | 0.196 | 0.298 | 0.512 | 0.090 | 0.020 | 0.060 |
| Aesthetics (AVA) | 0.589 | 0.609 | 0.644 | 0.566 | 0.700 | 0.710 | 0.750 |
| Aesthetics (KonIQ-10k) | 0.739 | 0.713 | 0.744 | 0.681 | 0.800 | 0.790 | 0.860 |
| Recency (HCI) | 0.600 | 0.592 | 0.631 | 0.571 | 0.780 | 0.770 | 0.820 |

strong, simple baseline that should be considered before applying such dedicated regression efforts. See Appendix E for detailed SOTA comparisons.

**CLIP embeddings are more rankable than non-CLIP embeddings.** We observe in Table 4 that on age, aesthetics, recency and pitch, the best CLIP encoder (*e.g.,* CLIP-ConvNeXt with an SRCC of 0.928 on Adience) wins out against the best non-CLIP encoder (*e.g.,* vanilla ConvNeXt with an SRCC of 0.871 on Adience). On crowd count, the best CLIP encoders are largely tied with the best non-CLIP encoders (*e.g.,* CLIP-RN50 at 0.870 vs vanilla RN50 at 0.864 for UCF-QNRF). On yaw and roll, DINO massively outperforms CLIP encoders (*e.g.,* 0.804 vs 0.440 for yaw). In conclusion, apart from isolated but interesting exceptions, CLIP encoders generally outperform or match non-CLIP encoders.

**Some attributes are better ranked than others.** For example, the average SRCC over the two age datasets is $\sim 0.8$ (see Table 3). Similar average SRCCs are observed for crowd count and pitch angle. Image aesthetics and recency are less well-ranked with average SRCCs between $0.65$ and $0.75$. Even within the same dataset (BIWI Kinect), yaw and roll angles with SRCCs of $0.434$ and $0.218$ are quite poorly ranked in comparison to pitch ($0.803$). We hypothesize that attribute-wise rankabilities are directly proportional to attribute-wise variety present in the training data. See also Appendix A.

**Caveats.** The current results are empirical, and our claims are based on the set of attributes considered in our study (we partially address this with a wider set of attributes in Appendix C). Despite following choices established in the literature, we may sometimes not uncover the most optimal finetuned upper bounds. A broader study involving theoretical support for rankability, using more ordinal attributes and possibly stronger upper bounds would be a promising direction for future work.

> **Takeaway from §3.** In general, visual embeddings are highly rankable compared to both the lower bound and upper bound baselines, although there exist variations across attributes (*e.g.,* most encoders struggle to rank based on yaw and roll angles) and encoders (*e.g.,* CLIP embeddings are, in general, more rankable than non-CLIP embeddings).

## 4 How to (efficiently) find the rank axis?

In Section 3, we showed that visual embeddings are generally rankable. However, the rank axes $v_A$, determining the direction along which the continuous attribute $A$ is sorted accordingly (Section 3.1), were learned using a lot of training data with continuous-attribute annotations that are typically expensive to collect. We examine whether rank axes can also be discovered in more sample- and label-efficient manners.

We organise the section by first tackling an easier setting with abundant data and then more challenging settings with less data available. We begin with the setting where we have a fraction of the dataset with continuous attribute labels (Section 4.1). Then, we examine the possibility to compute the rank axes with a few "extreme" points that require no cumbersome continuous attribute annotation (Section 4.2). For further practicality, we examine whether the obtained rank axes are resilient to domain shifts (Section 4.3). Finally, we discuss potential strategies to compute the rank axis in a zero-shot manner with text encoders in vision-language models (Section 4.4).

### 4.1 Learning using a fraction of the dataset with continuous attribute labels

In this section, we investigate the learnability of the rank axis $v_A$ for an attribute of interest $A$, when fewer samples with continuous attribute annotations are available. We report the results in Figure 2. We choose the datasets UTKFace, Adience and AVA because of their relatively large sizes compared to the other datasets used in our experiments.

**Only a fraction of training data is sufficient.** For age on the Adience dataset, only $\sim 1$k training samples out of over 11k are sufficient for covering as much as $95\%$ of the gap between the SRCCs of the no-train baseline and full-dataset linear regression. Similarly, for image aesthetics on AVA (CLIP-ViT), only $\sim 16$k (CLIP-ViT) or even $\sim 8$k (CLIP-ConvNeXt) data points out of $\sim 230$k are sufficient. It is evident that learning the rank axis often requires only a fraction of the original training dataset.

### 4.2 Learning using a small number of extreme pairs

We consider the practical scenario where a user wishes to sort their vector database using an arbitrary attribute $A$. They do not have access to continuous attribute annotations (as was assumed in Section 4.1), but can readily obtain a few samples at the "extremes" of the desired rank axis. We test the effectiveness of this approach.

**Setup.** Given training and test splits $\mathcal{X}_{\text{train}}$ and $\mathcal{X}_{\text{test}}$, respectively, we sample sets of images $S_l$ and $S_u$ from the lower and upper extremes of $\mathcal{X}_{\text{train}}$, respectively. This simulates the scenario where a user may obtain such "extreme" images using a readily available source like a Web search engine. Next, we

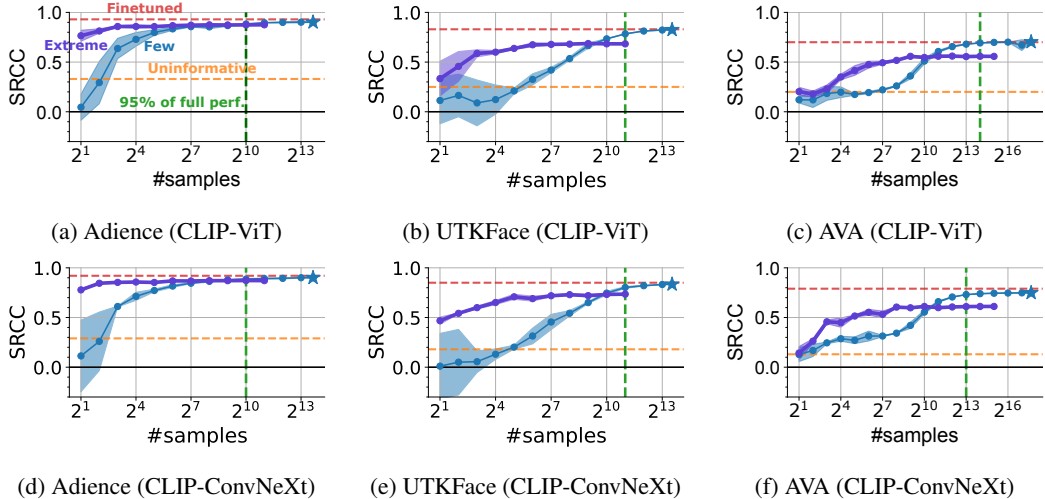

(a) Adience (CLIP-ViT)    (b) UTKFace (CLIP-ViT)    (c) AVA (CLIP-ViT)

(d) Adience (CLIP-ConvNeXt)    (e) UTKFace (CLIP-ConvNeXt)    (f) AVA (CLIP-ConvNeXt)

Figure 2: **Learning using a small number of samples with continuous labels vs extreme samples without continuous labels: extreme samples win out in the small-train-set regime. Extreme** refers to training using samples from the extreme ends of the ranking axis (without continuous labels), while **Few** refers to learning on continuously labelled samples. **Uninformative** refers to the no-train baseline while **Finetuned** refers to the finetuned upper bound. Dashed vertical lines labeled **95% of full perf.** indicate coverage of 95% of the gap between **Uninformative** and **Finetuned**. Full-dataset linear regression performance is denoted by a marker (★).

calculate the "lower extreme cluster" $x_l = \text{mean}(S_l)$ and "upper extreme cluster" $x_u = \text{mean}(S_u)$. Finally, the vector $v_A = \frac{x_u - x_l}{||x_u - x_l||}$ gives the rank axis (akin to a *steering vector* [54, 59]).

**Observations.** We report the results in Figure 2, comparing with the scenario where GT continuous attribute labels are available. We observe that when the size of the training dataset is extremely small (upto $\sim 1$k), the extreme-pairs method performs better than or at par with regular training using an equal number of continuously labelled samples. This effect can be seen most prominently in the Adience dataset where a rank axis obtained from just two extreme samples achieves an SRCC of $\sim 0.75$ on average, while the vanilla training case results in an average SRCC close to $0$. As the training dataset continues to grow in size, regular training with labelled data catches up and finally surpasses the extreme-pairs method. It is nevertheless striking that extreme pairs perform so well at lower sizes of the training dataset.

**Takeaway.** If one has an extremely small number of samples (1k or less), obtaining extreme samples from both ends of the rank axis (with no additional labeling) may constitute a better expenditure of resources than obtaining continuous attribute labels.

### 4.3   Robustness of rank axis

We now consider the case where there is no access to samples from the target distribution. Assuming that a rank axis was previously learned using some source distribution, how transferable is it to other distributions? We investigate this using three attributes: age, crowd count, and image aesthetics.

**Setup**. In our main experiments, we use two age datasets (UTKFace, Adience), three crowd count datasets (UCF-QNRF, ShanghaiTech-A, and ShanghaiTech-B), and two aesthetics datasets (AVA, KonIQ-10k). Within each set of datasets containing the same attribute, we then test how well a rank axis learned from one dataset transfers to another, and vice versa. We employ SRCC on the target dataset as our transferability metric in Table 5. Further, we also report cosine similarities between each pair of rank axes in Table 6. As reference points, we also report inter-attribute observations (*e.g.,* transfrability from an age dataset to an aesthetics dataset).

**Transfer is non-trivial and asymmetric.** For example, the rank axis learned from Adience has an SRCC of $0.680$ on UTKFace, which is significantly higher than the SRCCs of the same rank axis on other datasets. Given that the two datasets have different labeling systems (especially with Adience labels being much more coarse than those in UTKFace), this is not immediately intuitive, or trivial,

Table 5: **Rank axis transferability**. Spearman rank correlation coefficients when a rank axis is *trained* on dataset $i$ (rows) and *evaluated* on dataset $j$ (columns).

| | | | | **Evaluated on** | | | |
|---|---|---|---|---|---|---|---|
| | | Age (UTKFace) | Age (Adience) | Crowd (UCF-QNRF) | Crowd (ST-A) | Crowd (ST-B) | Aesthetics (AVA) | Aesthetics (KonIQ10k) |
| **Trained on** | Age (UTKFace) | +0.81 | +0.55 | −0.12 | −0.11 | −0.15 | −0.07 | +0.05 |
| | Age (Adience) | +0.68 | +0.91 | −0.13 | −0.10 | −0.21 | +0.01 | −0.08 |
| | Crowd (UCF-QNRF) | +0.16 | −0.39 | +0.87 | +0.82 | +0.66 | +0.01 | +0.07 |
| | Crowd (ST-A) | +0.05 | −0.14 | +0.73 | +0.75 | +0.72 | +0.13 | +0.07 |
| | Crowd (ST-B) | +0.31 | +0.17 | +0.41 | +0.39 | +0.86 | +0.11 | +0.05 |
| | Aesthetics (AVA) | −0.12 | −0.11 | +0.22 | +0.00 | +0.49 | +0.70 | +0.45 |
| | Aesthetics (KonIQ10k) | +0.06 | −0.08 | +0.03 | +0.18 | +0.05 | +0.28 | +0.79 |

and indicates the presence of some (albeit imperfect) "age" axis in the embedding space. At the same time, transfer in the opposite direction is not necessarily equally good (albeit still non-trivial), *i.e.,* a rank axis trained on UTKFace achieves an SRCC of only $0.55$ on Adience.

**Rank directions are non-trivially correlated.** For example, the cosine similarity between age axes trained on UTKFace and Adience is $0.360$. While this similarity is much lower than $1.0$, it is still significant given the high dimensionality of the embedding space. This observation again indicates the presence of a "universal" age axis which was (albeit not perfectly) captured by regressors trained on both datasets. Notably, some unexpected correlations also emerge, *e.g.,* age (UTKFace) and aesthetics (KonIQ-10k) rank axes have a cosine similarity of $0.220$. This suggests the presence of unintended correlations in the training / test data.

Table 6: **Cosine similarity of rank axes**. We compute geometric alignment of rank axes trained on dataset $i$ (rows) and dataset $j$ (columns).

| | | | | **Dataset j** | | | |
|---|---|---|---|---|---|---|---|
| | | Age (UTKFace) | Age (Adience) | Crowd (UCF-QNRF) | Crowd (ST-A) | Crowd (ST-B) | Aesthetics (AVA) | Aesthetics (KonIQ10k) |
| **Dataset i** | Age (UTKFace) | +1.00 | +0.36 | +0.14 | +0.08 | +0.06 | −0.04 | +0.22 |
| | Age (Adience) | +0.36 | +1.00 | +0.02 | +0.04 | +0.03 | +0.03 | +0.05 |
| | Crowd (UCF-QNRF) | +0.14 | +0.02 | +1.00 | +0.54 | +0.26 | +0.00 | +0.21 |
| | Crowd (ST-A) | +0.08 | +0.04 | +0.54 | +1.00 | +0.31 | +0.01 | +0.14 |
| | Crowd (ST-B) | +0.06 | +0.03 | +0.26 | +0.31 | +1.00 | +0.08 | +0.07 |
| | Aesthetics (AVA) | −0.04 | +0.03 | +0.00 | +0.01 | +0.08 | +1.00 | +0.29 |
| | Aesthetics (KonIQ10k) | +0.22 | +0.05 | +0.21 | +0.14 | +0.07 | +0.29 | +1.00 |

## 4.4 Zero-shot setting

For VLMs, language is a potentially data-free approach to finding a rank axis. In principle, a text prompt could correspond to a rank axis in the embedding space. The SRCC of our linear regressors trained in Section 3 then sets an upper bound to the SRCC of any rank axis recovered via prompting. In this section, we investigate the gap between this linear regression upper bound and zero-shot prompt search.

**Setup.** We consider two zero-shot settings. In the **single-prompt** setting, the direction is defined by the embedding of a text prompt (*e.g.,* "a picture of an old person"). In the **text-difference** setting, the direction is defined by the difference between the embeddings of two text prompts, each describing one extreme of the given attribute (*e.g.,* "a picture of an empty room" and "a picture of a crowded room"). The latter setting is similar to the one used in [66]. We perform a prompt search over 500

prompts generated by GPT-4o [45] for the single-prompt setting, and 100 prompt pairs generated similarly for the text-difference setting. We report our results in Table 7.

**Observations and takeaway.** Overall, zero-shot methods are suboptimal. For instance, even the best zero-shot result corresponds to $\rho = 0.782$ on Adience vs. the corresponding linear model at $\rho = 0.917$. The gap is more pronounced for some attributes (*e.g.,* , $\rho = 0.449$ vs. $\rho = 0.790$ for image recency). While more optimal prompts may have been overlooked during the search, this also reflects a realistic setting where one exhausts all intuitive prompt choices. Currently, it is evident that language-based prompting lags considerably behind linear regression using image data, although the text-difference method improves upon vanilla prompting.

Table 7: **Zero-shot results.**

| Attribute (Dataset) | Zero-shot One prompt | Zero-shot Difference | Linear upper bound |
|---|---|---|---|
| Age (UTKFace) | 0.577 | 0.600 | 0.817 |
| Age (Adience) | 0.670 | 0.782 | 0.917 |
| Crowd Count (UCF-QNRF) | 0.523 | 0.315 | 0.867 |
| Crowd Count (ST-A) | 0.487 | 0.242 | 0.763 |
| Crowd Count (ST-B) | 0.590 | 0.535 | 0.880 |
| Pitch (Kinect) | 0.520 | 0.617 | 0.887 |
| Yaw (Kinect) | 0.117 | 0.060 | 0.307 |
| Roll (Kinect) | -0.070 | -0.010 | 0.057 |
| Aesthetics (AVA) | 0.367 | 0.410 | 0.720 |
| Aesthetics (KonIQ-10k) | 0.103 | 0.547 | 0.817 |
| Recency (HCI) | 0.190 | 0.449 | 0.790 |

> **Takeaway from §4.** One may often efficiently discover the (almost) optimal rank axis using only a fraction of the total labelled data, or even pairs of extremes without continuous labels. Learned rank axes are quite transferable across different datasets, suggesting the presence of "universal" rank axes, although they may be somewhat spuriously entangled with the datasets used to obtain them.

# 5   Conclusion and future work

In this work, we investigate visual encoders for the presence of ordinal information. Extensive experiments reveal not only that such information is present, as indicated by the high Spearman rank correlation of nonlinear regressors, but also that most of the available ordinality is **linearly** encoded, as indicated by the small gap between the performances of linear and MLP regressors. The embedding space indeed possesses a "rankable" structure. This is unexpected and practically useful.

These findings provoke further questions. Most notably, the *linearity* of ordinal information suggests that one could potentially also characterise embeddings as interpretable collections of latent ordinal subspaces. This would involve discovering a far bigger set of ordinal attributes: we leave this exciting direction to future work.

**Broader impact statement.** Our work is largely a foundational analysis into the structure of embedding spaces and, as such, has no direct negative societal impacts. However, we highlight a few potential, indirect impacts. First, the ability to rank images containing personal or sensitive information (e.g. faces) with respect to arbitrary attributes may risk exposing certain individuals that are at the extreme ends of a spectrum (e.g. income level). Second, ordering individuals along a single attribute axis, such as gender, may reinforce existing stereotypes and offend and marginalize certain demographic groups.

**Acknowledgements.** This work was supported by the German Federal Ministry of Education and Research (BMBF): Tübingen AI Center, FKZ: 01IS18039A. The authors thank the International Max Planck Research School for Intelligent Systems (IMPRS-IS) for supporting Ankit Sonthalia and Arnas Uselis. The authors are also grateful to Wei-Hsiang Yu, the first author of [78], for helpful insights.

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

# Contents

# A   Why are vision embeddings rankable?

While theoretical analysis was outside of the scope of our study, we conduct a few further analyses to support our main claims.

## A.1   Attribute-wise variability and label noise

We conjecture that rankability along a given ordinal axis is directly related to variability of different attributes in the training data. The wider the attribute distribution, the stronger the optimization pressure may be for rankability to emerge.

**Background**. In our experiments using CLIP (obtained from timm [72], trained on LAION-400M [58]), we find that rankability for pitch (SRCC = 0.91) is higher compared to rankabilities for yaw (0.31) and roll (0.40). We set out to verify whether the variability of the head orientation angles in LAION-400M face images follows the same trend.

**Setup**. We randomly sample 100 faces in LAION-400M using a face detection model [43]. We use the same model to estimate yaw, pitch and roll angles of each face. We quantify the variability of each attribute in the LAION-400M dataset using the standard deviations of angles among the 100 faces.

**Results and discussion**. The standard deviations for pitch, roll and yaw are 21.2, 11.7 and 11.0 degrees, respectively, in LAION-400M. This concurs with the ranking of rankability among the three attributes: pitch > yaw > roll. This corroborates our hypothesis to a certain degree.

As for why CLIP models perform better, we conjecture that label noise present in captions may play a role. Below, we reason via an example.

When training a CLIP model on image-caption pairs that include age-related descriptors, there may often exist ambiguity in the use of age terms. For instance, middle-aged individuals may sometimes be described as "young", while young individuals are described as "middle-aged". However, one rarely describes young individuals as "old". This asymmetric labeling noise may implicitly encourage the model to embed images such that the representations of "young" and "middle-aged" individuals are closer to each other than either of them is to representations of "old" individuals, i.e.,

$$\mathrm{dist}(\mathrm{old}, \mathrm{young}) > \mathrm{dist}(\mathrm{middle\text{-}aged}, \mathrm{young}).$$

This may, in turn, indirectly produce the ordinal structure observed in our study.

## A.2   Why does DINO outperform CLIP on yaw and roll?

CLIP relies on language supervision. As long as the embedding space places similar image-text pairs together and dissimilar ones apart, there is limited incentive for fine-grained understanding of the image. We hypothesize that LAION-400M also has far fewer captions describing yaw and roll. Intuitively, for yaw, "looking to the left/right" can be ambiguous, and it is even more unnatural to describe roll using language. We verify this intuition empirically by counting the frequency of such phrases in LAION captions: out of approx. 13M captions, pitch is described in approx. 800k captions, yaw in approx. 5k captions while roll is only described in approx. 30 captions. Hence, while there is some incentive to learn pitch, there are reduced incentives for learning yaw and roll.

DINO relies on self-distillation for training, i.e., without language guidance. Additionally, the training process involves several augmentations. As such, fine grained understanding of the visual structure in the image becomes important, likely leading to enhanced coverage of attributes like yaw and roll.

## A.3   On which attributes would CLIP significantly outperform DINO?

We observe that CLIP significantly outperforms DINO [47] on image-level attributes like aesthetics and recency. We hypothesize that this can again be explained on the basis of attribute variability. CLIP was pretrained on a largely uncurated dataset (LAION) with very basic caption filters in place. This would have allowed varying qualities of images into CLIP's training pipeline, leading to high variation in quality. On the other hand, DINO's training dataset, LVD-142M, is much better curated, and likely contains only high quality images. Without variation in quality, DINO does not learn much about this attribute.

## B  Behavior of out-of-scope values

We design an experiment below for analysing *extrapolation* behaviour.

**Setup**. The UTKFace dataset split used by us (following prior work [78]) contains age labels between 21 and 60. We split the dataset into two subsets: one with ages 21-50 and the other with ages 51-60. We test whether the rank axis obtained over the 21-50 split applies to the 51-60 split by computing the SRCC against the ground-truth labels on the 51-60 split. As a control, we compare against another rank axis computed using the full age range 21-60.

**Results and conclusion**. We observe that the rank axis obtained on ages 21-50 achieves an SRCC of 0.174 on the 51-60 split, compared to 0.267 when trained on the full range 21-60. Our results suggest that extrapolation is indeed harder than interpolation.

## C  Results on more attributes

Broadly, our study focuses on four categories of attributes:

- Age progression.
- Geometric understanding (head pose).
- Scene complexity and counting (crowd count).
- Image quality (aesthetics, modernness).

Our reliance on continuously annotated attributes naturally constrains us to this selection. However, a qualitative exploration of other attributes is also interesting; we address this next.

### C.1  Ordinal attributes on dSprites

dSprites [38] provides a synthetic testbed for rankability along more "difficult" attributes like color and size.

**Setup**. We use pretrained CLIP ViT-B/32 embeddings to analyze the rankability of 5 attributes: hue, x-position, y-position, scale, and orientation. For hue, we consider the segment between **red** and **yellow** to avoid cyclicity of the attribute. We compare the no-train, linear, non-linear, and finetuned performances in Table A8.

Table A8: dSprites results.

| Attribute | No-train | Linear | Nonlinear | Finetuned |
|---|---|---|---|---|
| Hue | 0.919 | 0.920 | 0.996 | 1.000 |
| PosX | 0.551 | 0.877 | 0.966 | 0.998 |
| PosY | 0.706 | 0.966 | 0.993 | 0.999 |
| Scale | 0.788 | 0.980 | 0.985 | 0.986 |
| Orientation | 0.151 | 0.716 | 0.908 | 0.982 |

**Results and discussion**. Hue is already well-represented in untrained representations (no-train, 0.919), whereas CLIP pretraining does not seem to boost ordinality (linear, 0.920). We conjecture that hue is often not a discriminatory signal in caption-based contrastive training. Other attributes are significantly better represented (e.g. 0.151 no-train $\rightarrow$ 0.716 linear for orientation). For some attributes, frozen embeddings are as rankable as finetuned ones; for example, scale ordering using rank axis (0.980) vs dedicated fine-tuning (0.986). We conclude that CLIP embeddings are inherently capable of ranking these attributes.

### C.2  Qualitative study: "luxuriousness"

We study the rankability of CLIP with respect to "luxuriousness" using images from the "kitchen room & dining room table" class in Open Images V7 [28].

**Setup.** We randomly sample 20 pairs of images and ask three annotators (in this case, the authors themselves) to answer which image in the pair is more "luxurious". The "ground truth" luxuriousness is determined via majority vote. We measure quality of the rank axis by measuring accuracy of the prediction of the more luxurious image in each pair. The random chance baseline is 0.50 and the upper bound is given by average inter-annotator agreement. We repeat this process over 3 independent trials with different sets of 20 pairs.

**Obtaining a rank axis from visual extremes.** We prompt GPT-4o [45] to generate two synthetic reference images: one depicting a minimal dining room table setup, while another represents a visually luxurious version of the same scene. We compute the rank axis as the L2-normalized difference between their CLIP image embeddings.

**Results.** The model achieves an accuracy of $0.75 \pm 0.07$, well above the random chance accuracy of $0.50$ and at the same level as the average inter-annotator agreement of $0.73 \pm 0.10$. This suggests that CLIP encodes linear ordinal structure to support ranking along semantic attributes like luxuriousness. As shown in the main paper, the rank axis can easily be obtained using just two extreme samples.

# D   Further details on datasets and dataset-specific results

In the main paper, we present results for each dataset aggregated over all models (Table 2 in the main paper) and rankabilities (Spearman $\rho$) for each model-dataset pair (Table 3 in the main paper). While the main results convey the primary evidence towards our claim (vision embeddings are rankable), we also report more detailed results on each dataset in this section. Please also refer to Table 1 in the main paper for a condensed overview of all datasets considered.

## D.1   Age

**UTKFace**, introduced in [83], is a dataset of face images with age labels ranging from 0 to 116. Following [78, 27], we use a smaller subset with ages ranging between 21 and 60. The dataset was downloaded from the official website (https://susanqq.github.io/UTKFace/). We report results in Table A9.

Table A9: **UTKFace (Age)**. Spearman's rank correlation $\rho$ across evaluation strategies. **No-train**: linear probe on untrained encoder. **Rankability**: linear probe on pretrained encoder. **Nonlinear**: MLP on frozen encoder. **Finetuned**: encoder + head trained end-to-end. Higher is better.

| Model | No-train
lower bound | Rankability
main | Nonlinear
upper bound | Finetuned
upper bound |
|---|---|---|---|---|
| ResNet-50 | 0.212 | 0.633 | 0.636 | 0.762 |
| ViT-B/32 | 0.283 | 0.739 | 0.749 | 0.737 |
| ConvNeXtV2-L | 0.283 | 0.772 | 0.776 | 0.815 |
| DINOv2 ViT-B/14 | 0.054 | 0.770 | 0.770 | 0.812 |
| OpenAI CLIP ResNet-50 | 0.130 | 0.820 | 0.830 | 0.790 |
| OpenAI CLIP ViT-B/32 | 0.250 | 0.810 | 0.830 | 0.830 |
| OpenCLIP ConvNeXt-L (D, 320px) | 0.180 | 0.820 | 0.840 | 0.850 |
| **Mean** | 0.199 | 0.766 | 0.776 | 0.799 |

**Adience**, introduced in [11], is another age dataset. Unlike UTKFace, it contains coarse labels (8 age groups instead of exact ages). We use the "aligned" version of the images and five-fold cross-validation as in [78]. Results can be found in Table A10.

Table A10: **Adience (Age)**. Spearman's rank correlation $\rho$ across evaluation strategies.

| Model | No-train lower bound | Rankability main | Nonlinear upper bound | Finetuned upper bound |
|---|---|---|---|---|
| ResNet-50 | 0.328 | 0.723 | 0.759 | 0.894 |
| ViT-B/32 | 0.310 | 0.828 | 0.860 | 0.892 |
| ConvNeXtV2-L | 0.522 | 0.871 | 0.885 | 0.910 |
| DINOv2 ViT-B/14 | 0.120 | 0.853 | 0.877 | 0.914 |
| OpenAI CLIP ResNet-50 | 0.070 | 0.898 | 0.914 | 0.894 |
| OpenAI CLIP ViT-B/32 | 0.292 | 0.924 | 0.922 | 0.928 |
| OpenCLIP ConvNeXt-L (D, 320px) | 0.220 | 0.928 | 0.932 | 0.938 |
| **Mean** | 0.266 | 0.861 | 0.878 | 0.910 |

## D.2 Crowd count

**UCF-QNRF**, introduced in [7], is a large crowd counting dataset containing images from diverse parts of the world. We use the official download link at https://www.crcv.ucf.edu/data/ucf-qnrf/ and the official train-test splits. Results can be found in Table A11.

Table A11: **UCF-QNRF (Crowd Count)**. Spearman's rank correlation $\rho$ across evaluation strategies.

| Model | No-train lower bound | Rankability main | Nonlinear upper bound | Finetuned upper bound |
|---|---|---|---|---|
| ResNet-50 | 0.466 | 0.864 | 0.870 | 0.826 |
| ViT-B/32 | 0.288 | 0.837 | 0.840 | 0.794 |
| ConvNeXtV2-L | −0.054 | 0.810 | 0.816 | 0.938 |
| DINOv2 ViT-B/14 | 0.219 | 0.788 | 0.842 | 0.961 |
| OpenAI CLIP ResNet-50 | 0.240 | 0.870 | 0.880 | 0.820 |
| OpenAI CLIP ViT-B/32 | 0.280 | 0.870 | 0.870 | 0.900 |
| OpenCLIP ConvNeXt-L (D, 320px) | 0.100 | 0.860 | 0.860 | 0.960 |
| **Mean** | 0.220 | 0.843 | 0.854 | 0.886 |

**ShanghaiTech**, introduced in [82], is another crowd counting dataset consisting of two parts: A and B. While part A was crawled from the Internet and features larger crowds in general, part B was taken from metropolitan areas of Shanghai and features much smaller crowds. We use the DropBox link available at https://github.com/desenzhou/ShanghaiTechDataset and official train-test splits for both parts. Results can be found in Table A12 and Table A13.

Table A12: **ShanghaiTech-A (Crowd Count)**. Spearman's rank correlation $\rho$ across evaluation strategies.

| Model | No-train lower bound | Rankability main | Nonlinear upper bound | Finetuned upper bound |
|---|---|---|---|---|
| ResNet-50 | 0.359 | 0.799 | 0.802 | 0.529 |
| ViT-B/32 | 0.148 | 0.700 | 0.623 | 0.558 |
| ConvNeXtV2-L | 0.061 | 0.695 | 0.753 | 0.834 |
| DINOv2 ViT-B/14 | −0.017 | 0.653 | 0.722 | 0.786 |
| OpenAI CLIP ResNet-50 | 0.200 | 0.760 | 0.770 | 0.510 |
| OpenAI CLIP ViT-B/32 | 0.010 | 0.750 | 0.770 | 0.700 |
| OpenCLIP ConvNeXt-L (D, 320px) | 0.080 | 0.780 | 0.800 | 0.910 |
| **Mean** | 0.120 | 0.734 | 0.749 | 0.689 |

Table A13: **ShanghaiTech-B (Crowd Count)**. Spearman's rank correlation $\rho$ across evaluation strategies.

| Model | No-train lower bound | Rankability main | Nonlinear upper bound | Finetuned upper bound |
|---|---|---|---|---|
| ResNet-50 | 0.280 | 0.879 | 0.906 | 0.672 |
| ViT-B/32 | 0.225 | 0.878 | 0.889 | 0.710 |
| ConvNeXtV2-L | 0.070 | 0.867 | 0.876 | 0.955 |
| DINOv2 ViT-B/14 | 0.020 | 0.821 | 0.869 | 0.972 |
| OpenAI CLIP ResNet-50 | 0.270 | 0.890 | 0.900 | 0.690 |
| OpenAI CLIP ViT-B/32 | 0.040 | 0.860 | 0.860 | 0.900 |
| OpenCLIP ConvNeXt-L (D, 320px) | 0.040 | 0.890 | 0.910 | 0.980 |
| **Mean** | 0.135 | 0.869 | 0.887 | 0.840 |

## D.3   Headpose (Euler angles)

The **BIWI Kinect** dataset, introduced in [13], is a collection of 24 different videos wherein the subject of the video sits about a meter away from a Kinect (https://en.wikipedia.org/wiki/Kinect) sensor and rotates their head to span the entire range of possible head-pose angles pitch (rotation about the x-axis), yaw (rotation about the y-axis) and roll (rotation about the z-axis). As there exists no official split that we know of, we randomly hold out 6 sequences for testing. Results can be found in Table A14 (pitch), Table A15 (yaw) and Table A16 (roll).

Table A14: **Kinect (Pitch)**. Spearman's rank correlation $\rho$ across evaluation strategies.

| Model | No-train lower bound | Rankability main | Nonlinear upper bound | Finetuned upper bound |
|---|---|---|---|---|
| ResNet-50 | 0.359 | 0.663 | 0.615 | 0.973 |
| ViT-B/32 | 0.401 | 0.673 | 0.505 | 0.951 |
| ConvNeXtV2-L | 0.548 | 0.909 | 0.882 | 0.984 |
| DINOv2 ViT-B/14 | 0.231 | 0.716 | 0.986 | 0.979 |
| OpenAI CLIP ResNet-50 | 0.450 | 0.860 | 0.870 | 0.970 |
| OpenAI CLIP ViT-B/32 | 0.400 | 0.920 | 0.940 | 0.980 |
| OpenCLIP ConvNeXt-L (D, 320px) | 0.450 | 0.880 | 0.880 | 0.990 |
| **Mean** | 0.405 | 0.803 | 0.811 | 0.975 |

Table A15: **Kinect (Yaw)**. Spearman's rank correlation $\rho$ across evaluation strategies.

| Model | No-train lower bound | Rankability main | Nonlinear upper bound | Finetuned upper bound |
|---|---|---|---|---|
| ResNet-50 | 0.160 | 0.624 | 0.726 | 0.990 |
| ViT-B/32 | 0.209 | 0.305 | 0.305 | 0.838 |
| ConvNeXtV2-L | 0.113 | 0.384 | 0.716 | 0.989 |
| DINOv2 ViT-B/14 | −0.046 | 0.804 | 0.871 | 0.994 |
| OpenAI CLIP ResNet-50 | −0.060 | 0.120 | 0.530 | 0.980 |
| OpenAI CLIP ViT-B/32 | 0.160 | 0.360 | 0.330 | 0.990 |
| OpenCLIP ConvNeXt-L (D, 320px) | 0.010 | 0.440 | 0.700 | 0.990 |
| **Mean** | 0.078 | 0.434 | 0.597 | 0.967 |

Table A16: **Kinect (Roll)**. Spearman's rank correlation $\rho$ across evaluation strategies.

| Model | No-train lower bound | Rankability main | Nonlinear upper bound | Finetuned upper bound |
|---|---|---|---|---|
| ResNet-50 | 0.202 | 0.352 | 0.430 | 0.930 |
| ViT-B/32 | 0.098 | 0.196 | 0.375 | 0.477 |
| ConvNeXtV2-L | 0.550 | 0.298 | 0.368 | 0.963 |
| DINOv2 ViT-B/14 | 0.256 | 0.512 | 0.551 | 0.912 |
| OpenAI CLIP ResNet-50 | 0.140 | 0.090 | 0.300 | 0.920 |
| OpenAI CLIP ViT-B/32 | −0.060 | 0.020 | 0.170 | 0.850 |
| OpenCLIP ConvNeXt-L (D, 320px) | −0.130 | 0.060 | 0.090 | 0.960 |
| **Mean** | 0.151 | 0.218 | 0.326 | 0.859 |

## D.4 Aesthetics (Mean Opinion Score)

The **Aesthetics Visual Analysis (AVA)** dataset, introduced in [41], is a large-scale dataset including aesthetic preference scores provided by human annotators. Each image is labeled by multiple annotators, each assigning a score in the range 1-10. The mean opinion score (MOS) of the image is then computed as a weighted average over the ratings where the weight of a rating is provided by its frequency. We use the split provided by [78] in their official repository (https://github.com/uynaes/RankingAwareCLIP/tree/main/examples). Results are reported in Table A17.

Table A17: **AVA (Image Aesthetics)**. Spearman's rank correlation $\rho$ across evaluation strategies.

| Model | No-train lower bound | Rankability main | Nonlinear upper bound | Finetuned upper bound |
|---|---|---|---|---|
| ResNet-50 | 0.237 | 0.589 | 0.628 | 0.672 |
| ViT-B/32 | 0.157 | 0.609 | 0.666 | 0.672 |
| ConvNeXtV2-L | 0.158 | 0.644 | 0.685 | 0.728 |
| DINOv2 ViT-B/14 | 0.057 | 0.566 | 0.648 | 0.590 |
| OpenAI CLIP ResNet-50 | 0.150 | 0.700 | 0.710 | 0.700 |
| OpenAI CLIP ViT-B/32 | 0.200 | 0.710 | 0.730 | 0.700 |
| OpenCLIP ConvNeXt-L (D, 320px) | 0.130 | 0.750 | 0.780 | 0.790 |
| **Mean** | 0.156 | 0.653 | 0.692 | 0.693 |

**KonIQ-10k**, introduced in [17], is another aesthetics or image quality assessment (IQA) dataset that aims to model naturally occurring image distortions with mean opinion scores ranging roughly between 1 and 100. We use the official train-test splits. Results can be found in Table A18.

Table A18: **KonIQ-10k (Image Aesthetics)**. Spearman's rank correlation $\rho$ across evaluation strategies.

| Model | No-train lower bound | Rankability main | Nonlinear upper bound | Finetuned upper bound |
|---|---|---|---|---|
| ResNet-50 | 0.563 | 0.739 | 0.739 | 0.874 |
| ViT-B/32 | 0.488 | 0.713 | 0.753 | 0.813 |
| ConvNeXtV2-L | 0.487 | 0.744 | 0.765 | 0.930 |
| DINOv2 ViT-B/14 | 0.324 | 0.681 | 0.753 | 0.948 |
| OpenAI CLIP ResNet-50 | 0.400 | 0.800 | 0.840 | 0.900 |
| OpenAI CLIP ViT-B/32 | 0.460 | 0.790 | 0.830 | 0.890 |
| OpenCLIP ConvNeXt-L (D, 320px) | 0.320 | 0.860 | 0.870 | 0.950 |
| **Mean** | 0.435 | 0.761 | 0.793 | 0.901 |

## D.5   Image recency

**Historical Color Images (HCI)**, introduced in [49], was designed for the task of classifying an image by the decade during which it was taken. Therein emerges a natural ordering over the decades, defining the ordinal attribute of image "modernness" or "recency". We use the split provided by [78] in their repository (https://github.com/uynaes/RankingAwareCLIP/tree/main/examples) and report the results in Table A19.

Table A19: **HCI (Historical Color Images)**. Spearman's rank correlation $\rho$ across evaluation strategies.

| Model | No-train lower bound | Rankability main | Nonlinear upper bound | Finetuned upper bound |
|---|---|---|---|---|
| ResNet-50 | 0.351 | 0.600 | 0.592 | 0.614 |
| ViT-B/32 | 0.377 | 0.592 | 0.618 | 0.529 |
| ConvNeXtV2-L | 0.362 | 0.631 | 0.663 | 0.771 |
| DINOv2 ViT-B/14 | 0.131 | 0.571 | 0.601 | 0.748 |
| OpenAI CLIP ResNet-50 | 0.320 | 0.780 | 0.770 | 0.760 |
| OpenAI CLIP ViT-B/32 | 0.430 | 0.770 | 0.780 | 0.760 |
| OpenCLIP ConvNeXt-L (D, 320px) | 0.300 | 0.820 | 0.790 | 0.870 |
| **Mean** | 0.324 | 0.680 | 0.688 | 0.722 |

# E   Comparison with existing works

Our main aim in this work is to understand the rankability emerging out of the structure in visual embedding spaces, and we contextualize our numbers using reference metrics (lower bound provided by the no-encoder baseline and upper bound provided by nonlinear regression and finetuned encoders). This section additionally provides comparisons with prior work to further support our claims.

## E.1   Comparison with SOTA works

Prior research has presented results from dedicated or general efforts to solve the datasets considered in our study. Such works often require modification of the entire encoder or final embeddings, with dedicated loss functions or even architectural changes. This significantly limits adaptability. For each new attribute to rank a dataset, a user needs to employ a considerable amount of compute to train a dedicated embedding.

In Table A20, we also provide comparisons with such state-of-the-art results to further contextualize our results. Although architectural and training dataset differences mean that this comparison is not

always fair, we emphasize the contrast in implementational simplicity between dedicated efforts and simple linear regression over pretrained embeddings that are often readily available and easy to use. Our comparisons suggest that many existing off-the-shelf encoders contain sufficient information to rank datasets according to common attributes and simple linear regression over pretrained embeddings is a strong baseline that must be considered before developing dedicated regression methods (please see next page for Table A20).

Table A20: **Comparing linear / nonlinear regression against recent state-of-the-art methods.** "Linear" and "Nonlinear" use regression over CLIP-ConvNeXt-L embeddings. Dashes indicate metrics unreported in prior work. We take the numbers for age, aesthetics and recency from [78], and crowd count from [37]. Under "Downstream model", we report the components used on top of pretrained visual embeddings (CLIP or non-CLIP models); sometimes, we also report if the encoder itself was retrained. Under "Downstream data", we report the additional data used for training the method. Finally, under "Other ingredients", we also report miscellaneous extra components used by the corresponding method. All method interpretations are to the best of our knowledge.

| Attribute | Dataset | Method | Spearman $\rho$ | MAE | Downstream model | Downstream data | Other ingredients |
|---|---|---|---|---|---|---|---|
| Age | UTKFace | Yu et al. [78] | – | 3.83 | Cross-attn encoder, two ranking heads, learnable text prompt tokens | Images with age labels | Text encoder |
| | | MiVOLO [27] | – | 4.23 | Regression heads | Body images, face patches, age labels | Feature enhancer module for fused joint representations |
| | | Linear | – | 4.25 | Linear regressor | Images with age labels | None |
| | | Nonlinear | – | 4.10 | 2-layer MLP regressor | Images with age labels | None |
| Age | Adience | Yu et al. [78] | – | 0.36 (0.03) | Cross-attn encoder, two ranking heads, learnable text prompt tokens | Images with age labels | Text encoder |
| | | OrdinalCLIP [34] | – | 0.47 (0.06) | (Retrain image encoder for the task) | Images with age labels | Text encoder; learn "continuous" rank prototype (text) embeddings for each rank |
| | | Linear regressor | – | 0.48 (0.02) | Linear | Images with age labels | None |
| | | Nonlinear | – | 0.45 (0.02) | 2-layer MLP regressor | Images with age labels | None |
| Crowd Count | UCF-QNRF | CLIP-EBC [37] | – | 80.3 | Blockwise classification module | Images with count labels | Text encoder |
| | | CrowdCLIP [35] | – | 283.3 | (Retrain image encoder) | Crowd images | Text encoder, three-stage progressive filtering during inference |
| | | Linear | – | 246.4 | Linear | Images with count labels | None |
| | | Nonlinear | – | 248.0 | 2-layer MLP | Images with count labels | None |
| Crowd Count | ST-A | CLIP-EBC [37] | – | 52.5 | Blockwise classification module | Images with count labels | Text encoder |
| | | CrowdCLIP [35] | – | 146.1 | (Retrain image encoder) | Crowd images | Text encoder, three-stage progressive filtering during inference |
| | | Linear | – | 167.1 | Linear | Images with count labels | None |
| | | Nonlinear | – | 151.7 | 2-layer MLP | Images with count labels | None |
| Crowd Count | ST-B | CLIP-EBC [37] | – | 6.6 | Blockwise classification module | Images with count labels | Text encoder |
| | | CrowdCLIP [35] | – | 69.3 | (Retrain image encoder) | Crowd images | Text encoder, three-stage progressive filtering during inference |

| Attribute | Dataset | Method | Spearman $\rho$ | MAE | Downstream model | Downstream data | Other ingredients |
|---|---|---|---|---|---|---|---|
| | | Linear | – | 34.7 | Linear | Images with count labels | None |
| | | Nonlinear | – | 29.7 | 2-layer MLP | Images with count labels | None |
| Aesthetics | AVA | Yu et al. [78] | 0.747 | – | Cross-attn encoder, two ranking heads, learnable text prompt tokens | Images with MOS labels (1–10) | Text encoder |
| | | CLIP-IQA [67] | 0.415 | – | Softmax over two similarity scores | None | Text encoder, prompt engineering, remove position embedding |
| | | Linear | 0.749 | – | Linear | Images with MOS labels(1–10) | None |
| | | Nonlinear | 0.775 | – | 2-layer MLP | Images with MOS labels (1–10) | None |
| Aesthetics | KonIQ-10k | Yu et al. [78] | 0.911 | – | Cross-attn encoder, two ranking heads, learnable text prompt tokens | Images with MOS labels (1–100) | Text encoder |
| | | CLIP-IQA [67] | 0.727 | – | Softmax over two similarity scores | None | Text encoder, prompt engineering, remove position embedding |
| | | Linear | 0.860 | – | Linear | Images with MOS labelss (1–100) | None |
| | | Nonlinear | 0.870 | – | 2-layer MLP | Images with MOS labels (1–100) | None |
| Recency | HCI | Yu et al. [78] | – | 0.32 (0.03) | Cross-attn encoder, two ranking heads, learnable text prompt tokens | Images with decade labels | Text encoder |
| | | OrdinalCLIP [34] | – | 0.67 (0.03) | (Retrain image encoder for the task) | Images with decade labels | Text encoder; learn "continuous" rank prototype (text) embeddings for each rank |
| | | Linear | – | 0.64 | Linear | Images with decade labels | None |
| | | Nonlinear | – | 0.60 | 2-layer MLP | Images with decade labels | None |

## E.2 Is the finetuning upper bound good enough?

While we train our finetuned (upper bound) models using regular MSE loss, there exist specialised ranking loss functions introduced in prior works that we will refer to as GOL [29] and RnC [81]. While direct comparisons training the encoders used in our main paper using the loss functions from GOL and RnC were infeasible because of resource constraints, we confirm using a small ablation whether finetuning is a strong-enough upper bound for our analysis. We achieve this by comparing finetuning with MSE loss against the loss functions proposed in [29, 81].

Table A21: Comparison with GOL [29]

| Linear | Nonlinear | Finetuned | GOL |
|---|---|---|---|
| 7.06 | 6.23 | 4.88 | 4.35 |

Table A22: Comparison with RnC [81].

| Linear [ResNet18; DINO] | Nonlinear [ResNet18; DINO] | FT [ResNet18] | RnC |
|---|---|---|---|
| 9.97; 8.18 | 9.50; 7.76 | 6.57 | 6.14 |

**Observations and conclusions.**

We present our findings in Table A21 and Table A22. We observe that standard finetuning using MSE loss on downstream tasks (using the same or similar models) yields performance quite close to that of [29, 81]. On UTKFace and AgeDB [40], finetuned VGG16 and ResNet18 models achieve MAEs of 4.88 and 6.57, only modestly outperformed by [29] (4.35) and RnC [81] (6.14), respectively. For RnC, we use the ResNet18 model from `timm` while theirs was trained from scratch, however, for GOL, we used the same initial weights as [29].

We emphasize that our goal is complementary to GOL and RnC: while they propose methods for improving rank estimation, our study aims to evaluate rankability already present in off-the-shelf visual embeddings. Given our primary focus on probing intrinsic structure of pretrained embeddings using architecture-agnostic tools, and that finetuning itself already reduces the gap significantly, we conclude that finetuning provides a sufficiently strong upper bound for our analysis.

