# OpenReview forum: "On the rankability of visual embeddings"
_NeurIPS.cc/2025/Conference — NeurIPS 2025 poster_

### Official Review · Reviewer_SNnm · 2025-06-30

**Clarity:** 3
**Significance:** 2
**Originality:** 2
**Rating:** 4
**Confidence:** 4

**Summary:**

The paper analyzes whether existing visual encoders have rankable embeddings. The authors define that an embedding space is rankable, if there exists a rank axis such that the projection onto it preserves the ordering of a continuous attribute. They claim that embeddings of modern visual encoders are highly rankable compared to lower bound metrics provided by the untrained-encoder baseline, and upper bound metrics provided by nonlinear regression and finetuned encoders. The paper also finds that a small number of extreme samples often suffice to recover meaningful rank axes.

**Questions:**

- For the upper-bound settings, are the MLPs and encoders trained using just a regular MAE or MSE loss?

- It seems that the few-shot performance would vary significantly depending on how the instances were sampled. If random sampling was used, how many trials were run to obtain the reported results?

- Attributes like age have a large number of available datasets, such as MORPH II[1], CLAP2015[2], AgeDB[3], and CACD[4]. Would it be possible to conduct experiments on several of these additional datasets to further verify the rankability and rank axis transferability across a broader range of age datasets?

[1] Karl Ricanek and Tamirat Tesafaye. MORPH: A longitudinal image database of normal adult age-progression. In Proc. IEEE Int. Conf. on Automatic Face and Gesture Recognition, 2006.

[2] Sergio Escalera, Junior Fabian, Pablo Pardo, Xavier Baró, Jordi Gonzàlez, Hugo J. Escalante, Dusan Misevic, Ulrich Steiner, and Isabelle Guyon. ChaLearn looking at people 2015: Apparent age and cultural event recognition datasets and results. In Proc. ICCV Workshops, 2015.

[3] Stylianos Moschoglou, Athanasios Papaioannou, Christos Sagonas, Jiankang Deng, Irene Kotsia, and Stefanos Zafeiriou. AgeDB: The first manually collected, in-the-wild age database. In Proc. CVPR, 2017.

[4] Bor-Chun Chen, Chu-Song Chen, and Winston H Hsu. Face recognition and retrieval using cross-age reference coding with cross-age celebrity dataset. IEEE Trans. Multimedia, 2015.

**Ethical Concerns:**

["NO or VERY MINOR ethics concerns only"]

**Final Justification:**

Many of my initial concerns have been addressed. Given the interesting topic and its potential for diverse applications, I have decided to raise my score.

**Limitations:**

Yes

**Paper Formatting Concerns:**

No concerns.

**Quality:**

3

**Strengths And Weaknesses:**

**Strengths**
- The proposed algorithm in this paper demonstrates a reasonable formulation for analyzing whether existing encoders capture ordinal information.
- The paper supports its claims through extensive evaluations over a diverse range of datasets and model architectures.
- It also covers few-shot and zero-shot learning scenarios, showing that rank axes can be effectively recovered using only a small number of extreme samples.

**Weaknesses**
- Some important details are missing, such as how the nonlinear and fine-tuned methods were trained.
- The paper lacks sufficient references to existing literature on methods for constructing ordinal embeddings. Are the performance gaps between linear regression over trained embeddings still small when models are trained using the loss functions proposed in [1], [2]?
- For an embedding to be practically useful in real-world ranking tasks, it must have a high degree of ordinality. Given that there are existing encoders specifically trained to produce strong ordinal embeddings, it is unclear whether analyzing or relying on the ordinality of a generic encoder serves a meaningful purpose.
- The paper lacks in-depth analysis on why certain attributes yield lower SRCCs. It instead relies on general, unsupported hypotheses(L183-188).
- The number of datasets per attribute was limited to a maximum of 2~3. Given the abundance of datasets available for attributes such as age and aesthetics, a more extensive evaluation across a broader range of datasets would have further substantiated the generalization capability and rank axis transferability proposed in this paper.


[1] Seon-Ho Lee, Nyeong-Ho Shin, and Chang-Su Kim. Geometric order learning for rank estimation. In Proc. NeurIPS, 2022.

[2] Kaiwen Zha, Peng Cao, Jeany Son, Yuzhe Yang, and Dina Katabi. Rank-N-Contrast: Learning continuous representations for regression. In Proc. NeurIPS, 2023.

---

> ### Author Rebuttal · Authors · 2025-07-31
>
> We appreciate the reviewer’s detailed comments and insightful questions. We address the remaining concerns below.
>
> ## W1: How were the nonlinear and finetuned methods trained?
>
> We provide the training details in §3.2.3 below.
>
> **Nonlinear regression.** We use a 2-layer MLP with 128 hidden dimensions and ReLU non-linearity (we will add this part). We randomly sample 30 hyperparameter configurations from a search space defined as follows. Learning rate and weight decay are sampled log-uniformly from ranges \[10^{-6}, 10^{-1}\] and \[10^{-7}, 10^{-4}\] respectively. Horizontal flipping augmentation is likewise switched on or off randomly. The learning rate is decayed to zero over the course of training using a cosine annealing schedule. We train for 100 epochs with a batch size of 128\.
>
> **Finetuning.** For ResNet50 and ConvNeXt, we conduct a grid search over 2 learning rates {10^{-4}, 10^{-3}}, and 2 weight decays {0, 10^{-5}}. For the other models, we conduct a wider grid search over 3 LRs {10^{-7}, 10^{-6}, 10^{-5}}, 4 weight decays {0, 10^{-5}, 10^{-4}, 10^{-3}}. The downstream learning rate is always fixed at 0.1, and we always use horizontal flip augmentations. We train all models for 20 epochs. Batch size was fixed at 128 except for ConvNeXt models and DINOv2 where it was reduced to 16\.
>
> Please let us know if further clarification is needed.
>
> ## W2.1: Lacking references to existing literature on methods for constructing ordinal embeddings.
>
> We discuss existing methods for constructing ordinal embeddings \[57, 40, 20, 10\] in §2. In essence, these works are application-driven, aiming to modify CLIP’s embeddings or the encoder itself for specific ordinal regression tasks while we study whether *general* vision encoders already possess rankability with respect to *arbitrary* attributes. We thank the reviewer for mentioning \[1, 2\] below. As with prior methods discussed above, \[1,2\] train attribute-specific encoders (VGG16, ResNet18, and so on) for solving specific ordinal regression tasks. In contrast, our work aims to understand the linear, ordinal structure *already present* in the embedding space.
>
> Other interesting future applications include:
>
> - **Zero-shot control in generative models**: Rank axes from the encoder can transfer to latent control in generation tasks, offering an interpretable way to steer outputs along semantic attributes.
>
> - **Model auditing and diagnosis**: When a model fails to predict certain continuous attributes, our method may provide an additional debugging tool for the intermediate representations to understand better why and how it fails.
>
> We will add the discussion in §2.
>
> \[1\] Lee et al., Geometric order learning for rank estimation. NeurIPS, 2022\.
> \[2\] Zha et al., Rank-N-Contrast: Learning continuous representations for regression. NeurIPS, 2023\.
>
> ## W2.2: Are the performance gaps between linear regression over trained embeddings still small when models are trained using the loss functions proposed in \[1\], \[2\]?
>
> This is a good point. While direct comparisons training the encoders used in our main paper using the loss functions from GOL and RnC were infeasible because of resource constraints, we evaluated regression over the models used in the original studies. Our interpretation of the reviewer's question is whether finetuning is a strong-enough upper bound for our analysis (please correct us if this is not the case); our goal is to evaluate this by comparing finetuning with MSE loss against the loss functions proposed in \[1, 2\].
>
> **Results on UTKFace using VGG16**.
>
> | Linear | Nonlinear | Finetuned | GOL |
> | :---- | :---- | :---- | :---- |
> | 7.06 | 6.23 | 4.88 | 4.35 |
>
> **Results on AgeDB**.
>
> | Linear \[ResNet18; DINO\] | Nonlinear \[ResNet18; DINO\] | Finetuned \[ResNet18\] | Rank-N-Contrast |
> | :---- | :---- | :---- | :---- |
> | 9.97; 8.18 | 9.50; 7.76 | 6.57 | 6.14 |
>
> **Observations and conclusions.**
>
> We observe that standard finetuning using MSE loss on downstream tasks (using the same or similar models,) yields performance quite close to that of \[1, 2\]. On UTKFace and AgeDB, finetuned VGG16 and ResNet18 models achieve MAEs of 4.88 and 6.57, only modestly outperformed by \[1\] (4.35) and RnC \[2\] (6.14), respectively. For RnC \[2\], we use the ResNet18 model from `timm` while theirs was trained from scratch, however, for GOL \[1\], we used the same initial weights as \[1\].
>
> We emphasize that our goal is complementary to GOL and RnC: while they propose methods for improving rank estimation, our study aims to evaluate rankability already present in off-the-shelf visual embeddings. Given our primary focus on probing intrinsic structure of pretrained embeddings using architecture-agnostic tools, and that finetuning itself already reduces the gap significantly, we conclude that finetuning provides a sufficiently strong upper bound for our analysis. We again thank the reviewer for mentioning these works, and will discuss them in our paper.
>
> ## W3: does analysing ordinality of a generic encoder serve a real purpose given that dedicated ordinal encoders exist?
>
> Yes, studying the presence of ordinality in off-the-shelf encoders has significant practical implications.
>
> Current solutions for ordinal embeddings \[57, 40, 20, 10\] require modification of the entire encoder or final embeddings, with dedicated loss functions or even architectural changes. This significantly limits adaptability. For each new attribute to rank a dataset, a user needs to employ a considerable amount of compute to train a dedicated embedding.
>
> Our findings suggest that such a procedure is unnecessary. Many existing off-the-shelf encoders contain sufficient information to rank datasets according to common attributes. In fact, our approach to find a linear rank axis in frozen embeddings yields comparable performances against state-of-the-art embeddings designed to rank each attribute of interest \[Table A12\].
>
> We will make this clearer in the revision.
>
> ## W4: Lacking in-depth analysis on why certain attributes yield low rankability; unsupported hypotheses.
>
> We hypothesize that rankability with respect to a given attribute depends on the variability of that attribute present in the training data (lines 189-193 in the submission). We present an empirical case study over the head-pose attributes using CLIP and its training dataset, LAION-400M.
>
> **Background.** In our experiments using CLIP (obtained from `timm`, trained on LAION-400M), we find that rankability for pitch (SRCC \= 0.91) is higher compared to rankabilities for yaw (0.31) and roll (0.40). We set out to verify whether the variability of the head orientation angles in LAION-400M face images follows the same trend.
>
> **Setup.** We randomly sample 100 faces in LAION-400M using a face detection model \[1A\]. We use the same model to estimate yaw, pitch and roll angles of each face. We quantify the variability of each attribute in LAION-400M dataset using the standard deviations of angles among the 100 faces.
>
> **Results and discussion.** The standard deviations for pitch, roll and yaw are 21.2, 11.7 and 11.0 degrees, respectively, in LAION-400M. This concurs with the ranking of rankability among the three attributes: pitch \> yaw \> roll. This corroborates our hypothesis to a certain degree.
>
> \[1A\] Narayan, Kartik, et al. "Facexformer: A unified transformer for facial analysis." *arXiv* (2024).
>
>
> ## W5: Number of datasets per attribute was limited to 2-3; Evaluation on more datasets?
>
> Given practical limits in computational capacity, we strived to hit the right balance between the breadth of model choices (7 models), number of attributes (7 attributes), and number of datasets per attribute (2-3 datasets per attribute where available), which already surmounts to 150 GPU days of compute (§3.2.4).
>
> At the same time, we agree that additional datasets will further support our claims. We present additional results using CLIP-ViT-B/32 on the datasets referenced by the reviewer. We will include the full results in the camera-ready version.
>
> We confirm the finding in our work that rankability is observed for the suggested datasets too.
>
> |  | No-train | Linear | Nonlinear | Finetuning |
> | :---- | :---- | :---- | :---- | :---- |
> | AgeDB | 0.470    | 0.895   | 0.899    | 0.906 |
> | MORPH II | 0.496    | 0.920    | 0.926    | 0.954 |
> | CLAP2015 | 0.078    | 0.934    | 0.944    | 0.944 |
> | CACD | 0.148    | 0.768   | 0.771 | 0.702 |
>
> Below we also report the **transfer results**.  For every age dataset we calculate two metrics:
>
> * **SRCC of rank axis obtained using the same dataset**. *We train on the dataset itself, then test on a held-out split from that same dataset.*  This is the first numeric column in the table.
>
> * **Mean transfer SRCC from other datasets**. We *train five separate rank-axis models, each on one of the other datasets; evaluate all five on the current dataset and take their average SRCC.*  This corresponds to the second column.
>
> For brevity we show only this averaged “Mean transfer SRCC from other datasets” in the table; the full per-source breakdown will be included in the camera-ready version.
>
> As in the main paper, we confirm non-trivial transfer of rank axes across age datasets.
>
> | Dataset | SRCC of rank axis obtained using the same dataset | Mean transfer SRCC from other datasets  |
> | :---- | :---- | :---- |
> | UTKFace | 0.814 | 0.712 |
> | Adience | 0.912 | 0.686 |
> | AgeDB | 0.895 | 0.706 |
> | MORPH II | 0.920 | 0.822 |
> | CLAP2015 | 0.934 | 0.820 |
> | CACD | 0.771 | 0.672 |
>
> ## Questions
>
> **Q1: Were upper-bound models trained using MAE or MSE loss?**
>
> We used the MSE loss across all settings.
>
> **Q2: How many trials were used to report the few-shot results?**
>
> We conducted three random trials. We plot the standard deviations in our results – the variation across trials gets smaller as we approach the “95% of full-training performance” point.
>
> **Q3: Experiments on additional datasets.**
>
> See W5 above.

---

> > ### Comment · Reviewer_SNnm · 2025-08-05
> >
> > Thank you for your response. I appreciate the additional experiments and the detailed clarifications. Most of my concerns have been addressed, so I will increase my score.

---

> > > ### Author Response · Authors · 2025-08-08
> > >
> > > Dear Reviewer SNnm,
> > >
> > > Thank you for your recognition of our work, and we are glad that our rebuttal addressed your concerns. Your insights and proposed experiments: especially, the ones including additional datasets, have further enriched our manuscript. We will make sure to add the content of our rebuttal to the camera-ready version.

---

### Official Review · Reviewer_nvX2 · 2025-07-02

**Clarity:** 2
**Significance:** 2
**Originality:** 2
**Rating:** 5
**Confidence:** 3

**Summary:**

The paper shows that image embeddings from common vision encoders can be ranked for specific attributes; e.g., for person counting or image quality. The method proposes different ways of learning such a rankable embedding by relying on linear or non-linear models to be trained on a small percentage of the dataset. Evaluation on a variety of tasks and attributes reveal that such rankable properties exist and that some attributes are easier to rank than others.

**Questions:**

I would be happy to increase my score following discussions on the linearity of the embedding, few-shot phrasing, and comparisons with very similar methods in the field of image quality.

**Ethical Concerns:**

["NO or VERY MINOR ethics concerns only"]

**Final Justification:**

The rebuttal was help to support claims in the paper, and clarify limitations of the method.

**Limitations:**

The paper has not discussed limitations or potential negative societal impact.

**Paper Formatting Concerns:**

N.A.

**Quality:**

2

**Strengths And Weaknesses:**

**Strengths**

*S1. The idea of learning rankable embeddings in common vision encoders is simple and effective.*
* This reveals that common encoders have this information and it can be learned easily with a low percentage of the training set.

*S2. The paper does not require fancy components, is applicable to a variety of vision encoders and tasks.*
* It is appreciated that the paper assess a variety of vision encoders and tasks, showing its wide applicability.

**Weaknesses**

*W1. The claim of linear embedding is unverified*
* The evaluation only focuses on measuring the Spearman rank correlation coefficient. What about the Pearson linear correlation coefficient?
* It would have been interesting to also provide qualitative or quantitative analysis on algorithmic operations in the embedding space. Does moving by the same magnitude and direction yield the same effect? (e.g., the number of people in the image always increases by N with this vector)

*W2. Few-shot claims are over-stated*
* While it is interesting to find attribute embeddings from existing models, the method still requires a thousand of examples! It seems like 5-10% of the dataset is still needed to learn the attribute embeddings. This is in contrast with the few-shot literature where few-shot usually means less than 10 examples. I would suggest to rephrase "few-shot", or mentions of "a small number of samples" to something more accurate.

*W3. Behavior of out-of-scope values is unclear*
* The method requires to provide extreme values to learn how to rank. Let say we present examples ranging from 0 to 100. What happens if at inference we present an image where the value is 200?

*W4. Comparisons are missing.*
* The evaluation only focuses on the selected vision encoders on various ranking tasks. But, what about the SOTA methods in each of these respective tasks? It would have been interesting to know what is the upper-band.
* Furthermore, the idea of ranking attributes had been recently explored in image quality assessment, where for example related work (e.g., [55], among many others) find a linear embedding between what is a "bad photo" and a "good photo". Direct comparisons with such methods is possible and would benefit the paper.

---

> ### Author Rebuttal · Authors · 2025-07-31
>
> We thank the reviewer for the careful reading and constructive questions. We hope that the additional experiments and clarifications below address the remaining concerns.
>
> ## W1: The claim of linear embedding is unverified.
>
> ### 1. What about Pearson linear correlation coefficient (PLCC)?
>
> We wish to clarify two related but distinct notions of linearity in the context of ordinal attribute embeddings:
>
> - Linearity of the rank axis refers to whether there exists a consistent direction in the embedding space along which an ordinal attribute (e.g. age, brightness) increases. Our work focuses on identifying this direction. To evaluate this, we compute Spearman’s rank correlation coefficient (SRCC) between the true attribute ranks and the projections of embeddings onto the identified rank axis.
>
> - Linearity of the ordering along the rank axis concerns whether items are spaced in a regular, linear fashion along this direction. For example, if the true attribute values increase by a fixed amount, do the corresponding embedding projections also show equal spacing? This reflects the regularity or uniformity of the embedding distribution along the rank axis.
>
> While our primary focus is on (1), we agree that further analysis of (2) could reveal useful insights into how ordinal attributes are represented in embedding space.
>
> We report PLCC  for the CLIP-ViT-B/32 embedding against 5 attributes below. For most attributes, the PLCC measures are close to SRCC (0.921 vs 0.932 for head pitch). This indicates that the ordering of attributes along the rank axis is largely linear. This is an interesting further insight into the structure of the embeddings. We will include the PLCC versions of Tables 2 & 3 in the paper and discuss the implications. We thank the reviewer for the additional insight.
>
> | Attribute                    | Dataset      | SRCC  | PLCC  |
> |-----------------------------|--------------|-------|-------|
> | Age                         | UTKFace      | 0.814 | 0.848 |
> | Aesthetics (Mean Opinion Score) | KonIQ-10k    | 0.793 | 0.818 |
> | Crowd count                 | UCF-QNRF     | 0.870 | 0.817 |
> | Head pitch                  | BIWI Kinect  | 0.921 | 0.932 |
> | Head yaw                    | BIWI Kinect  | 0.360 | 0.370 |
> | Head roll                   | BIWI Kinect  | 0.020 | -0.030 |
> | Image recency               | HCI          | 0.770 | 0.760 |
>
> ### 2. Algorithmic operations in the embedding space: does moving by the same magnitude and direction yield the same effect?
>
> Yes, to some degree. The Pearson's correlation results above show this indirectly. We show this more directly with a dedicated experiment below.
>
> **Setup**. We measure the calibration of the rank-axis projected distances of the embeddings against the ground-truth ordinal attribute. We use CLIP-ViT-B/32 embeddings on the ShanghaiTech-B and UTKFace datasets for the age and crowd count attributes, respectively. We then measure the linearity of the relationship between the ground-truth age and the projected distances using the coefficient of determination ($R^2$).
>
> **Results and conclusion**. The $R^2$ values are 0.761 for crowd count and 0.849 for age. This indicates a largely linear agreement between projected distance along the rank axis and the ground-truth attribute values.
>
> ## W2: Few-shot claims are over-stated.
>
> We thank the reviewer for pointing this out. We will revise all mentions of “few-shot” in the paper to a more accurate wording like “sub-1k supervision” or “5-10% of the dataset” to avoid confusion and false expectations.
>
> ## W3: Behavior of out-of-scope values: what happens if the inference image is outside the range of training images?
>
> We thank the reviewer again for a very interesting question. This will shed further light on the structural regularity of the embedding space.  We design an experiment below for the extrapolation behaviour.
>
> **Setup**. The UTKFace dataset contains age labels between 21 and 60. We split the dataset into two subsets: one with ages 21-50 and the other with ages 51-60. We test whether the rank axis obtained over the 21-50 split applies to the 51-60 split by computing the SRCC against the ground-truth labels on the 51-60 split. As a control, we compare against another rank axis computed by the full age range 21-60.
>
> **Results and discussion**. We observe that the rank axis obtained on ages 21-50 achieves an SRCC of 0.174 on the 51-60 set, compared to 0.267 when trained on the full range 21-60. Extrapolation is indeed harder than interpolation. It is thus recommended that users obtain extreme examples to cover the full dynamic range of attributes.
>
> ## W4: Comparisons are missing.
>
> ### 1. What about SOTA methods?
>
> We already have comparisons against the state-of-the-art models in Table A12 of the supplementary material. We observe that
> On UTKFace (age estimation), our rank axis over fixed embeddings achieves an MAE of 4.25, slightly underperforming [1] (MAE = 3.83) and performing on par with [2] (MAE = 4.23). This is remarkable given that [1, 2] rely on further modifications to the embeddings designed to enhance ordinality.  On UCF-QNRF (crowd-counting), likewise, our approach achieves an MAE of 246.4, even outperforming CrowdCLIP [3] a model designed for crowd counting.The results further support our thesis that pretrained vision embeddings are highly rankable, even compared to more complex specific systems designed for the prediction of respective attributes. We will include this discussion in the main paper.
>
> ### 2. Comparisons with image quality assessment works like CLIP-IQA [55] that discuss similar ideas.
>
> We thank the reviewer for mentioning CLIP-IQA. In Table A12 of the supplementary material, we include comparisons with CLIP-IQA. On KonIQ-10k (aesthetics), our linear rank axis achieves an SRCC of 0.860, outperforming CLIP-IQA (0.727). We emphasise that this comparison is not entirely direct: CLIP-IQA does not use supervised training data and relies solely on prompt-based scoring. However, we agree that such comparisons provide further contextualization of our work. We will include more of such quantitative comparisons in the main paper, instead of supplementary materials.
>
> ## Limitations: the paper has not discussed limitations or potential negative social impact.
>
> While we discuss the limitations in lines 189-193, we agree that we should discuss the potential negative societal impacts of our work. We recognize a few potential issues. First, the ability to rank images containing personal or sensitive information (e.g. faces)   with respect to arbitrary attributes may risk exposing certain individuals that are at the extreme ends of a spectrum (e.g. income level). Second, ordering individuals along a single attribute axis, such as gender, may reinforce existing stereotypes and offend and marginalize certain demographic groups. This discussion will be added to the manuscript.

---

> ### Author Response · Authors · 2025-08-06
>
> Dear Reviewer nvX2,
>
> Thank you for your earlier note indicating that you would consider increasing your score following discussions on the concerns raised in your review.
>
> We have provided a detailed response to address these points, but have not yet heard back from you. As we are now late in the discussion phase, we would appreciate knowing whether our clarifications have resolved your concerns and whether you are considering an increase in your score, or if there are any remaining issues we should address while time still allows.
>
> Best regards,
>
> Authors

---

> > ### Comment · Reviewer_nvX2 · 2025-08-06
> >
> > Thank you for the rebuttal, it is helpful. In particular, the PLCC experiments support the claims about linearity and I appreciate the further discussion about the limitations around extrapolation. I will increase my score.

---

> > > ### Author Response · Authors · 2025-08-08
> > >
> > > Dear Reviewer nvX2,
> > >
> > > Thank you for your appreciation of our rebuttal. Your insights and proposed experiments surrounding linearity and extrapolation have likewise added meaningfully to our manuscript. We will make sure to incorporate them into the camera-ready version.

---

### Official Review · Reviewer_pn5U · 2025-07-03

**Clarity:** 4
**Significance:** 3
**Originality:** 4
**Rating:** 5
**Confidence:** 3

**Summary:**

The paper investigates whether visual encoders (like CLIP model) inherently encode ordinal attributes along linear directions in the embedding space, an ability which the authors name as rankability. Through extensive evaluation on multiple models across multiple attribute datasets, the authors state that existing visual embedding spaces are indeed rankable, and often times the rank axis of ordinal attribute can be estimated using only 2 extreme samples (one sample possessing the least of the attribute, one sample possessing the most). In addition, the authors also propose few-shot and zero-shot learning methods to estimate this rank axis, as well as analyze how well a rank axis is transferrable across different datasets (within the same attribute). In overall, this paper shows a different research direction into understanding the structure of visual embedding space (in addition to existing work that study hierarchies, disentanglement, separability of embedding space).

**Questions:**

Questions presented in weaknesses section above, which are mostly around:
- Expand attribute diversity.
- Downstream applications of rankability. Is rankable a good thing?
- Should we train visual encoders to have good rankability in its embedding space? How can we train for it?

**Ethical Concerns:**

["NO or VERY MINOR ethics concerns only"]

**Final Justification:**

Thanks to the authors' rebuttal which includes a lot of high quality and useful discussion into the understanding of visual embedding space, I increase the Quality score from 3 to 4, while maintaining the original Rating as 5.

**Limitations:**

- Yes, the authors did discuss the limitation of the work.
- The authors state that their work does not have societal impacts. However, I believe it does to some extent. Discovering rankability of embedding space, as well as proposing zero-shot / few-shot to discover rank axis can make it become a lot easier to develop ranking feature for image database where we can rank images in terms of attributes that discriminate against certain groups of people.

**Quality:**

4

**Strengths And Weaknesses:**

Strengths:
- Good and clear writing.
- Estimating rankability is a clearly defined, new problem for understanding embedding space that has not been explored.
- Evaluation is done thoroughly across multiple models and attribute datasets, although the selected attributes are still a bit not diverse yet.
- It’s interesting to learn that discovering rank axis may sometimes require only 2 extreme samples, or even as few as 10% of the training data. Using only text prompts from VLMs to discover rank axis is also an interesting idea.

Weaknesses:
1/ Limited attributes and domain diversity: attributes are face-related (age) or low-level (yaw, pitch, roll, aesthetics), so generalizability is still questionable. It would be better to also study this on more difficult attributes (could be as a qualitative study only) such as color, size, or semantic attributes such as “luxurious”. To generate datasets for more diverse attributes, one way is to prompt state-of-the-art text-to-image generators to generate object images with increasing order of attributes.

2/ L187, the authors state one hypothesis on rankabiliy could be dependent on attribute-wise variety in the training data, but did not provide more details. This is interesting and is also worth studying in the paper, at least for the attributes with low rankability such as yaw and roll. In addition, is there any insight into why DINO performs so much better on yaw and roll than CLIP-based models? What do you think will be the attributes that CLIP-based model performs much better than DINO?

3/ Applicability: one potential application of the proposed approach is fast ranking in image database with respect to an attribute, but the authors did not discuss anything else. Apart from that, is there any other application? If we learn that a particular visual embedding space has high rankability, is that a good thing? Should we train visual encoders to have good rankability? If model A has better rankability than model B in terms of attribute X, would A perform better than B on classification task of attribute X? Is it necessary to study rankability?

---

> ### Author Rebuttal · Authors · 2025-07-31
>
> We sincerely thank the reviewer for their thoughtful and encouraging feedback. We are especially glad that the reviewer finds the framing of rankability as a property for understanding embedding spaces novel and well-motivated. We appreciate the suggestions for expanding attribute diversity and applications. We address each point below.
>
> ## W1: Attribute diversity
>
> 1\. *Attributes are face-related or low-level; generalizability is questionable*.
>
> We appreciate this observation. Broadly, our study focuses on four categories of attributes:
>
> - Age progression.
> - Geometric understanding (head pose).
> - Scene complexity and counting (crowd count).
> - Image quality (aesthetics, modernness).
>
> Our reliance on continuously annotated attributes naturally constrains us to this selection. However, as the reviewer notes in the following section, a qualitative exploration of other attributes is also interesting; we address this next.
>
> *2\. Qualitative study on more difficult attributes such as color, size, and semantic attributes like "luxurious"*.
>
> These experiments would indeed add meaningfully to the empirical breadth; we thank the reviewer for proposing them. Below, we present the results.
>
> - ### Ordinal attributes on dSprites: color, scale, position, orientation
>
> dSprites provides a synthetic testbed for rankability along more "difficult" attributes like color and size.
>
> **Setup.** We use pretrained **CLIP ViT-B/32 embeddings** to analyze the rankability of **5** attributes: **hue**, **x-position**, **y-position**, **scale**, and **orientation**. For hue, we consider the segment between red and yellow to avoid cyclicity of the attribute. We compare the no-train, linear, non-linear, and finetuned performances.
>
> | Attribute | No-train | Linear | Nonlinear | Finetuned |
> | :---- | :---- | :---- | :---- | :---- |
> | Hue | 0.919   | 0.920  |  0.996   | 1.000 |
> | PosX | 0.551 | 0.877 | 0.966 | 0.998 |
> | PosY | 0.706 | 0.966 | 0.993 | 0.999 |
> | Scale | 0.788 | 0.980 | 0.985 | 0.986 |
> | Orientation | 0.151 | 0.716 | 0.908 | 0.982 |
>
> **Results and discussion.** The table above shows the results. Hue is already well-represented in untrained representations (no-train, 0.919), whereas CLIP pretraining does not seem to boost ordinality (linear, 0.920). We conjecture that hue is often not a discriminatory signal in caption-based contrastive training. Other attributes are significantly better represented (e.g. 0.151 no-train $\\rightarrow$ 0.716 linear for orientation). For some, frozen embeddings are as rankable as finetuned ones; for example, scale ordering using rank axis (0.980) vs dedicated fine-tuning (0.986). CLIP embeddings are inherently capable of ranking these attributes.
>
> - ### Qualitative study: “Luxuriousness”
>
> We study the rankability of CLIP with respect to "luxuriousness" using images from the `kitchen room & dining room table` class in Open Images V7.
>
> **Setup.** We randomly sample 20 pairs of images and ask three annotators to answer which image in the pair is more "luxurious".  The "ground truth" luxuriousness is determined via majority vote. We measure quality of the rank axis by measuring accuracy of the prediction of the more luxurious image in each pair. The random chance baseline is 0.50 and the upper bound is given by average inter-annotator agreement. We repeat this process over 3 independent trials with different sets of 20 pairs.
>
> **Obtaining a rank axis from visual extremes.** We prompt GPT-4o to generate two synthetic reference images: one depicting a minimal dining room table setup, while another represents a visually luxurious version of the same scene. We compute the rank axis as the L2-normalized difference between their CLIP image embeddings.
>
> **Results.** The model achieves an accuracy of 0.75 ± 0.07, well above the random chance accuracy of 0.50 and at the same level as the average inter-annotator agreement of 0.73 ± 0.10. CLIP encodes linear ordinal structure to support ranking along semantic attributes like luxuriousness. As shown in the main paper, the rank axis can easily be obtained using just two extreme samples.
>
> ## W2: Impact of attribute-wise variety in the training data; why does DINO outperform CLIP on yaw and roll; on which attributes would CLIP significantly outperform DINO?
>
> We thank the reviewer for highlighting this direction. As a preliminary test for our hypothesis, we conducted a case study using headpose angles on CLIP.
>
> - ### Case study on headpose angles with CLIP
>
> We hypothesize that rankability w.r.t. a given attribute depends on its variability in training data (lines 189-193). We present an empirical study over head-pose attributes using CLIP and its training dataset, LAION-400M.
>
> **Background.** In our experiments using CLIP, we find that rankability for pitch (SRCC \= 0.91) is higher compared to yaw (0.31) and roll (0.40). We verify whether the variability of the head orientation angles in LAION-400M face images follows this trend.
>
> **Setup.** We randomly sample 100 faces in LAION-400M using a face detection model \[1A\] and the same model to estimate yaw, pitch and roll angles of each face. We quantify variability of each attribute using standard deviations among the 100 faces.
>
> **Results.** The standard deviations for pitch, roll and yaw are 21.2, 11.7 and 11.0 degrees, respectively, in LAION-400M. This concurs with the ranking of rankabilities: pitch \> roll \> yaw. This corroborates our hypothesis to a certain degree.
>
> \[1A\] Narayan, Kartik, et al. "Facexformer: A unified transformer for facial analysis." *arXiv* (2024).
>
> - ### CLIP vs DINO
>
> **DINO outperforms CLIP on yaw and roll.** CLIP relies on language supervision. As long as the embedding space places similar image-text pairs together and dissimilar ones apart, there is limited incentive for fine-grained understanding of the image. We hypothesize that LAION-400M also has far fewer captions describing yaw and roll. Intuitively, for yaw, “looking to the left/right” can be ambiguous, and it is even more unnatural to describe roll using language. We verify this intuition empirically by counting the frequency of such phrases in LAION captions: out of ~13M captions, pitch is described in \~800k captions, yaw in \~5k captions while roll is only described in \~30 captions.
>
> Hence, while there is some incentive to learn pitch, there are reduced incentives for learning yaw and roll.
>
> DINO relies on self-distillation for training, i.e., without language guidance. Additionally, the training process involves several augmentations. As such, fine grained understanding of the visual structure in the image becomes important, likely leading to enhanced coverage of attributes like yaw and roll.
>
> **CLIP outperforms DINO on image aesthetics and recency.** We observe that CLIP significantly outperforms DINO on image-level attributes like aesthetics and recency. We hypothesize that this can again be explained on the basis of attribute variability. CLIP was pretrained on a largely uncurated dataset (LAION) with very basic caption filters in place. This would have allowed varying qualities of images into CLIP’s training pipeline, leading to high variation in quality. On the other hand, DINO’s training dataset, LVD-142M, is much better curated, and likely contains only high quality images. Without variation in quality, DINO does not learn much about this attribute.
>
> ## W3: Applicability
>
> *1\. Is there any other application other than fast database sorting?* *Is rankability a good thing and should we train visual encoders to have it?*
>
> Higher rankability, by definition implies we may perform continuous regression without changing the underlying embedding. Current works on ordinal embeddings (\[57, 40, 20, 10\] among others), make dedicated efforts to refine the embedding space for ranking-awareness with respect to one attribute at a time. After applying the ranking-aware finetuning, it is often also unclear whether the same model can then continue to support other downstream tasks.
>
> In contrast, our work studies whether *existing* vision encoders *linearly* encode *general* ranking-awareness (i.e., with respect to several attributes) even without dedicated optimisation pressures or post-training modifications. Indeed, we find that vanilla CLIP embeddings, for instance, are already ordinal with respect to many attributes studied in prior works.
>
> Strikingly, simple linear regression over frozen embeddings often performs on par with these dedicated ordinal embeddings (we report detailed SOTA comparisons in Table A12 of the supplement).
>
> We believe that this observation in itself is surprising and practically valuable, as it eases all the ordinal regression tasks that would otherwise be handled by dedicated ordinal encoders.
>
> \[3A\] Yu et al. "Ranking-aware adapter for text-driven image ordering with CLIP." *ICLR 2025\.*
>
> *2\. If model A outperforms model B on rankability over attribute X, will the trend transfer over to classifying on the basis of X?*
>
> We expect that the existence of a rank axis should also help classification. Below, we conduct a preliminary experiment. As expected, the better regression model is also the better classification model.
>
> | Model | Regression (SRCC) | Classification (Accuracy) |
> | :---- | :---- | :---- |
> | ResNet50 | 0.63 | 0.48 |
> | CLIP-ViT-B/32 | 0.81 | 0.60 |
>
> **Q1: Expand attribute diversity.**
> See W1 above.
>
> **Q2: Downstream applications; is rankability good?**
> See W3 above.
>
> **Q3: Training vision encoders for rankability**
>
> *1\. Should we train for rankability?*
>
> See W3 above.
>
> *2\. How can we train for rankability?*
>
> Our study finds that rankability naturally emerges in many vision encoders, especially modern ones (e.g., CLIP). However, given the benefits of rankability, future work may also explore ways to induce it explicitly, perhaps, by including pairwise comparison annotations (image 1 \> image 2 for a given attribute), and adding a hinge loss to the pretraining objective.

---

> > ### Comment · Reviewer_pn5U · 2025-08-07
> >
> > Thank you for the authors’ response which is very detailed and data-driven. The response has satisfactorily addressed my major questions, with data and experiments to address my concerns. This includes: how DINO’s self-supervision with strong augmentation favor fine-grained understanding of visual structure; how CLIP’s training dataset implicitply allows for better rankability in terms of image aesthetics; the relative difficulty of pitch, roll and yaw are due to the variability of them in the training set; how we do not need dedicated optimisation (that hurts the embedding space) but still able to achieve rankability; etc. I would suggest the authors include these useful discussion in the main paper or appendix.

---

### Official Review · Reviewer_kX7E · 2025-07-03

**Clarity:** 4
**Significance:** 3
**Originality:** 3
**Rating:** 5
**Confidence:** 5

**Summary:**

This paper presents an intriguing experimental study that delves into the rankability of visual embeddings in great detail. It begins with concise mathematical definitions, followed by extensive experiments conducted from multiple perspectives: the input side (using different datasets), the output side (considering various ranking attributes), and the research subjects (different model types and specific models). The study finds that, under the paper's settings, visual features often exhibit good linear rankability, and CLIP-based models generally outperform non-CLIP models. Additionally, the paper explores efficient methods for obtaining projection directions under different settings.

**Questions:**

1. Regarding Weakness 1, we strongly recommend that the authors add relevant theoretical analysis to enhance the theoretical depth of the paper.​
2. Regarding Weakness 2.A, an in-depth exploration of the reasons for the poor performance of the Finetuned method compared to the Nonlinear method in specific scenarios is expected.​
3. Regarding Weakness 2.B, a more comprehensive summary of the impact of different factors on rankability is desired.​
4. Regarding Weakness 3, experiments on challenging datasets to verify the effectiveness of the method for obtaining projection directions are suggested.

**Ethical Concerns:**

["NO or VERY MINOR ethics concerns only"]

**Final Justification:**

The authors have addressed all our questions adequately in their rebuttal. This paper investigates a highly interesting phenomenon and offers valuable insights to the research field. Therefore, we propose upgrading the decision to **Accept**.

**Limitations:**

Yes

**Paper Formatting Concerns:**

No formatting concern.

**Quality:**

3

**Strengths And Weaknesses:**

## Strengths
1. **Clarity of Writing:** The paper is written with a highly logical and smooth flow, making it very accessible to readers. The conclusions are presented clearly and straightforwardly.​
2. **Comprehensive Experiments:** The paper conducts cross-research on multiple datasets, a wide range of attributes, and various models. The experimental results are thus relatively reliable.​
3. **Rigorousness in Writing:** The authors clearly define the applicable boundaries of the experimental conclusions in each section and analyze the results meticulously and rigorously.

## Weaknesses​
1. **Theoretical Analysis:** Although the paper provides extensive experimental research and analysis, it would be beneficial if the authors could offer some theoretical explanations. For instance, why do these visual embeddings possess good linear rankability? And why do CLIP models typically perform better in this regard? Such analyses would help summarize more universal and generalizable fundamental principles.​
2. **Experimental Results:**
    -  **A. Exceptional Cases**:​ In Table 2, we observe that in some scenarios, such as Crowd Count (ST-A) and Crowd Count (ST-B), the finetuned results are even worse than those of the Nonlinear method. Does this indicate potential optimization difficulties for these two methods? Or are there other underlying reasons?​
    - **B. Conclusion Summarization**: The authors mainly summarize the differences in rankability between different model types (CLIP and non-CLIP). It would be valuable if they could further explore and summarize how different dataset types, different attribute types, and even the loss functions used in model training affect rankability.​
3. **Method Design:** In Section 4, the paper analyzes methods for efficiently obtaining projection directions. However, in Figure 2, only datasets with good rankability performance are presented. We are curious whether the proposed efficient methods for obtaining projection directions remain effective on more challenging datasets, such as the Kinect dataset shown in Table 2.

---

> ### Author Rebuttal · Authors · 2025-07-31
>
> We thank the reviewer for their recognition of our findings and for their commendation of our experimental setup. Below, we address the remaining concerns.
>
> ## W1: Theoretical explanations: why do visual embeddings possess good linear rankability, and why do CLIP models typically perform better in this regard?
>
> While theoretical analysis was outside of the scope of our study, we conjecture that rankability along a given ordinal axis is directly related to variability of different attributes in the training data (lines 189-193). The wider the attribute distribution, the stronger the optimization pressure may be for rankability to emerge. To confirm this, we conduct a case study using CLIP.
>
> **Background.** In our experiments using CLIP (obtained from `timm`, trained on LAION-400M), we find that rankability for pitch (SRCC \= 0.91) is higher compared to rankabilities for yaw (0.31) and roll (0.40). We set out to verify whether the variability of the head orientation angles in LAION-400M face images follows the same trend.
>
> **Setup.** We randomly sample 100 faces in LAION-400M using a face detection model \[1A\]. We use the same model to estimate yaw, pitch and roll angles of each face. We quantify the variability of each attribute in the LAION-400M dataset using the standard deviations of angles among the 100 faces.
>
> **Results and discussion.** The standard deviations for pitch, roll and yaw are 21.2, 11.7 and 11.0 degrees, respectively, in LAION-400M. This concurs with the ranking of rankability among the three attributes: pitch \> yaw \> roll. This corroborates our hypothesis to a certain degree.
>
> \[1A\] Narayan, Kartik, et al. "Facexformer: A unified transformer for facial analysis." *arXiv* (2024).
>
> As for why CLIP models perform better, we conjecture that label noise present in captions may play a role. Below, we reason via an example.
>
> When training a CLIP model on image-caption pairs that include age-related descriptors, there may often exist ambiguity in the use of age terms. For instance, middle-aged individuals may sometimes be described as “young”, while young individuals are described as “middle-aged.” However, one rarely describes young individuals as “old.” This asymmetric labeling noise may implicitly encourage the model to embed images such that the representations of “young” and “middle-aged” individuals are closer to each other than either is to that of “old” individuals, i.e.,
>
> $$
> \text{dist}(\text{old}, \text{young}) > \text{dist}(\text{middle-aged}, \text{young}).
> $$
>
> This may, in turn, indirectly produce the ordinal structure observed in our study.
>
> ---
>
> ## W2: Experimental results
>
> A. *Finetuned results for some datasets like ST-A and ST-B are even worse than their Nonlinear counterparts.*
>
> This is a valid concern. In principle, finetuning should outperform regression on a frozen encoder. However, finetuning is highly sensitive to hyperparameter choices.
>
> That said, for smaller datasets such as ST-A and ST-B, the restricted hyperparameter range may lead to overfitting during finetuning. As a sanity check, we conduct a wider hyperparameter search (four times as long as the original search) for CLIP-RN50 on ST-A and ST-B, with results presented below (model selection using validation SRCC). We observe that with a wide enough parameter search, finetuning does outperform linear and nonlinear regression, albeit still not drastically (\~0.06 SRCC in each case). For most datasets, finetuning slightly outperforms linear and nonlinear regression by default; while the current sanity check indicates that even longer hyperparameter searches would not have altered our conclusions drastically.
>
> | Dataset | No-train | Linear | Nonlinear  | Finetuning (12 runs) | Finetuning (48 runs) |
> | :---- | :---- | :---- | :---- | :---- | :---- |
> | ST-A | 0.20 | 0.76 | 0.77 | 0.51 | 0.83 |
> | ST-B | 0.27 | 0.89 | 0.90 | 0.69 | 0.96 |
>
> ---
> B. *How do different dataset types, attribute types and training loss functions affect rankability?*
>
> We believe that rankability emerges out of the interaction between the attribute, the evaluation dataset, and the training objective.
>
> **Effect of attribute and dataset.** While we observe non-trivial rankabilities for all attributes and datasets (Table 2 in the paper), some patterns emerge. Age, crowd count and pitch angle are ranked better than image aesthetics and recency, while pitch is ranked significantly better than yaw and roll (lines 183-187). We validate our hypothesis around this observation in W1 above. In addition, we also observe that within the same attribute (age), Adience is ranked better (0.723) than UTKFace (0.623). We believe that this stems from the fact that labels are coarser in Adience with 8 age group labels instead of exact year as in UTKFace. We also invite the reviewer’s feedback on
>
> - our analysis on other attributes under Reviewer pn5U’s W1, and
> - our evaluations on other datasets under Reviewer SNnm’s W5.
>
> **Effect of training objective.** We consider 7 pretrained vision encoders. CLIP encoders were trained using the standard InfoNCE loss. Among the remaining encoders: DINOv2 was trained using self-distillation loss, ConvNeXtv2 using self-supervised FCMAE loss, and RN50 and ViTB32 were trained using supervised classification.
>
> - We observe that CLIP is, in general, more rankable than or at par with non-CLIP encoders on all attributes except for yaw and roll where DINO performs best (lines 175-182). We observe a mixed, attribute-dependent trend among these four encoders.
> - On age, both self-supervised models DINO and CNX perform on par with each other while outperforming the fully supervised models. On crowd count, pitch, aesthetics and image recency, CNX outperforms DINO to varying extents.
> - The fully supervised models win out over the self-supervised models on crowd count, while underperforming on all other attributes (when the best models out of both groups are considered).
>
> In conclusion, self-supervised training objectives seem to lead to better rankability than supervised ones, while CLIP's training outperforms both. Future studies may also consider the role of pre-training datasets.
>
> ---
> ## W3: Method design: do efficient methods for finding rank axes generalize to more difficult datasets like Kinect?
>
> To check this, we conducted experiments using Kinect (pitch) on CLIP. We report results in the table below. We observe again that using just 512 samples (\~5% of the full dataset), we already recover an SRCC of 0.9, i.e., more than 95% of the SRCC obtained upon training on the full training set.
>
> | \#shots | Test SRCC on pitch |
> | :---- | :---- |
> | 2 | 0.357 ± 0.132 |
> | 32 | 0.510 ± 0.090 |
> | 512  | 0.900 ± 0.040 |
> | 2048 | 0.934 ± 0.007 |
> | 9924 (full dataset) | 0.941 ± 0.000 |
> ---
> ## Questions
>
> **Q1: Theoretical analysis**
>
> See W1 above.
>
> **Q2: Why does Finetuned sometimes perform worse than Nonlinear?**
>
> See W2.A above.
>
> **Q3: Impact of different factors on rankability**
>
> See W2.B above.
>
> **Q4: Effectiveness of efficient rank-axis finding methods on more challenging datasets**
>
> See W3 above.

---

> > ### Comment · Reviewer_kX7E · 2025-08-02
> >
> > Thank you for your response. It has largely addressed our concerns, and we will consider raising our evaluation score.

---

> > > ### Author Response · Authors · 2025-08-02
> > >
> > > We thank Reviewer kX7E for their continued recognition of our work and for indicating the possibility of raising their score. We remain open to addressing any further questions or concerns.

---

### Note · Authors · 2025-08-16

We would like to thank the reviewers for their time and efforts spent reading our paper, and for adding valuable insights.

We are pleased that the paper was unanimously well-received, and that our rebuttal addressed the remaining concerns, with three reviewers (kX7E, nvX2, SNnm) indicating willingness to raise their scores. Notably,

- Reviewer kX7E found the paper’s flow “highly logical and smooth,” raised interesting questions about the factors that affect rankability, and encouraged us to extend our efficient rank-axis-finding experiments (§4) to the Kinect dataset.

- Reviewer pn5U recognized rankability as a “clearly defined, new problem for understanding embedding space”. The reviewer's questions motivated us to broaden the scope of attributes considered, including a qualitative attribute ("luxuriousness").

- Reviewer nvX2 appreciated the applicability of our findings to “a variety of vision encoders”. The reviewer questioned the regularity of the embedding space, motivating further experiments in that regard.

- Reviewer SNnm commended the evaluation as “extensive [...] over a diverse range of datasets and model architectures”. Following the reviewer's recommendation, we incorporated additional datasets to make our claims on rankability and rank-axis transferability even stronger.

We believe our work will inspire further valuable research on the understanding of embedding spaces. We are excited to incorporate the reviewers’ suggestions (addressed in detail in the respective rebuttals) into the camera-ready version of our paper. Upon acceptance, we will also include a link to the code repository for replicating our findings.

---

### Decision · Program_Chairs · 2025-09-17

**Decision:**

Accept (poster)

**Comment:**

The paper investigated if embeddings learned by current deep encoders have ordinal information. The answer turns out to be positive, and mapping between the embeddings and ordinal information is not difficult to do. This investigation is quite in depth, the task interesting and not sufficiently well explored. While reviewers initially had issues with some of the claims, they were fullyy ocnvinced by the rebuttal and all recommend acceptance, most without reservation. The AC recommends acceptance.